# Multi-year effect of wetting on CH$_4$ flux at taiga-tundra boundary in northeastern Siberia deduced from stable isotope ratios of CH$_4$

Ryo Shingubara[1,a], Atsuko Sugimoto[2,3,4], Jun Murase[5], Go Iwahana[2,6], Shunsuke Tei[2,3], Maochang Liang[1,b], Shinya Takano[1], Tomoki Morozumi[1], Trofim C. Maximov[7,8]

[1]Graduate School of Environmental Science, Hokkaido University, Sapporo, 060-0810, Japan
[2]Arctic Research Center, Hokkaido University, Sapporo, 001-0021, Japan.
[3]Faculty of Environmental Earth Science, Hokkaido University, Sapporo, 060-0810, Japan.
[4]Global Station for Arctic Research, Global Institution for Collaborative Research and Education, Hokkaido University, Sapporo, 060-0808, Japan.
[5]Graduate School of Bioagricultural Sciences, Nagoya University, Nagoya, 464-8601, Japan.
[6]International Arctic Research Center, University of Alaska Fairbanks, Fairbanks, 99775-7340, USA.
[7]Institute for Biological Problems of Cryolithozone, Siberian Branch, Russian Academy of Sciences, Yakutsk, 677890, Russia.
[8]Institute of Natural Sciences, North-Eastern Federal University, Yakutsk, 677000, Russia.
[a]Currently at: Graduate School of Environmental Studies, Nagoya University, Nagoya, 464-8601, Japan.
[b]Currently at: College of Horticulture and Gardening, Yangtze University, Jingzhou, 434023, China.

*Correspondence to*: Atsuko Sugimoto (atsukos@ees.hokudai.ac.jp)

**Abstract.** The response of $CH_4$ emission from natural wetlands due to meteorological conditions is important because of its strong greenhouse effect. To understand the relationship between $CH_4$ flux and wetting, we observed interannual variations in chamber $CH_4$ flux, as well as the concentration, $\delta^{13}C$, and $\delta D$ of dissolved $CH_4$ during the summer from 2009 to 2013 at the taiga-tundra boundary in the vicinity of Chokurdakh (70° 37′ N, 147° 55′ E), located on the lowlands of the Indigirka River in northeastern Siberia. We also conducted soil incubation experiments to interpret $\delta^{13}C$ and $\delta D$ of dissolved $CH_4$ and to investigate variations in $CH_4$ production and oxidation processes. Methane flux showed large interannual variations in wet areas of sphagnum mosses and sedges (36–140 mg $CH_4$ m$^{-2}$ day$^{-1}$ emitted). Increased $CH_4$ emission was recorded in the summer of 2011 when a wetting event with extreme precipitation occurred. Although water level decreased from 2011 to 2013, $CH_4$ emission remained relatively high in 2012, and increased further in 2013. Thaw depth became deeper from 2011 to 2013, which may partly explain the increase in $CH_4$ emission. Moreover, dissolved $CH_4$ concentration rose sharply by one order of magnitude from 2011 to 2012, and increased further from 2012 to 2013. Large variations in $\delta^{13}C$ and $\delta D$ of dissolved $CH_4$ were observed in 2011, and smaller variations were seen in 2012 and 2013, suggesting both enhancement of $CH_4$ production and less significance of $CH_4$ oxidation relative to the larger pool of dissolved $CH_4$. These multi-year effects of wetting on $CH_4$ dynamics may have been caused by continued soil reduction across multiple years following the wetting. Delayed activation of acetoclastic methanogenesis following soil reduction could also have contributed to the enhancement of $CH_4$ production. These processes suggest that duration of water saturation in the active layer can be important for predicting $CH_4$ emission following a wetting event in permafrost ecosystem.

## 1 Introduction

Atmospheric $CH_4$ has an important greenhouse effect (Myhre et al., 2013). The largest source of atmospheric $CH_4$ is the emission from natural wetlands, which is considered to be the main driver of interannual variations in the global $CH_4$ emission, depending on meteorological conditions such as air temperature and precipitation (Ciais et al., 2013). For instance, Dlugokencky et al. (2009) reported that high temperatures in the Arctic and high precipitation in the tropics led to high $CH_4$ emissions from natural wetlands, which caused the observed large growth rates in atmospheric $CH_4$ concentration during 2007 and 2008. Atmospheric $CH_4$ has been increasing from 2007 through the present (Nisbet et al., 2014).

Methane flux from wetland soil to the atmosphere (we define a positive flux value as $CH_4$ emission) is determined by three processes: $CH_4$ production, oxidation, and transport (Lai, 2009). Methane is produced by strictly anaerobic *Archaea* (methanogens) mainly via hydrogenotrophic methanogenesis ($4H_2 + CO_2 \rightarrow CH_4 + 2H_2O$) or acetoclastic methanogenesis ($CH_3COOH \rightarrow CH_4 + CO_2$) as an end product of organic matter decomposition (Lai, 2009). In the soil's aerobic zone, $CH_4$ is oxidized to $CO_2$ by methanotrophic bacteria utilizing $O_2$, which reduces $CH_4$ emission to the atmosphere (Lai, 2009). Underground $CH_4$ is transported to the atmosphere via bubble ebullition, diffusion through soil layers and surface water, and via aerenchyma of vascular plants (Lai, 2009).

High water levels can lead to development of reducing conditions in soil, which can promote $CH_4$ production or depress $CH_4$ oxidation, both leading to increases in $CH_4$ flux (Lai, 2009). This is reflected in the widely observed positive relationship between water level and $CH_4$ flux, found in a meta-analysis across the circum-Arctic permafrost zone (Olefeldt et al., 2013). Meanwhile, Desyatkin et al. (2014) observed increases in $CH_4$ flux during the second consecutive year of flooding at a thermokarst depression in boreal eastern Siberia. Treat et al. (2007) reported observations at a temperate fen in the northeastern USA showing that high water level coincided with high $CH_4$ flux in interannual variations. However, water level correlated negatively with $CH_4$ flux over shorter timescales, namely as monthly means or individual measurements. These observational results imply that wetting is not directly related to $CH_4$ flux in wetlands. To understand the relationship between wetting and $CH_4$ flux, it is necessary to assess the underlying processes.

Stable isotopes of $CH_4$ have been used to estimate production pathways of $CH_4$ (Sugimoto and Wada, 1993; Sugimoto and Wada, 1995; McCalley et al., 2014; Itoh et al., 2015), determine the fraction of oxidized $CH_4$ versus produced $CH_4$ (Marik et al., 2002; Preuss et al., 2013) and to study mechanisms of $CH_4$ transport by plants (Chanton, 2005). When $CH_4$ in soil is lost by oxidation or diffusion, both $\delta^{13}C$ and $\delta D$ of the remaining $CH_4$ increase. While the hydrogen isotope ratio increases more than that of carbon during oxidation, both ratios are considered to change to the same extent during diffusion. Thus it is useful to analyze both carbon and hydrogen isotopes of $CH_4$ to distinguish the effects of both of these processes (Chanton, 2005).

The taiga-tundra boundary ecosystem (or transition zone) contains vegetation types of both taiga and tundra ecosystems. Liang et al. (2014) reported that the distribution of vegetation types at the taiga-tundra boundary on the lowland of the Indigirka River in northeastern Siberia is controlled by soil moisture, which corresponds to microtopography. Larches, the dominant tree species in the taiga forests of eastern Siberia, grow on micro-reliefs with higher ground level and drier soil, while wetland vegetation such as sphagnum mosses and sedges, typically seen in wet tundra (van Huissteden et al., 2005; van der Molen et al., 2007), dominates lower and wetter micro-reliefs. Thus, it is reasonable to assume that the taiga-tundra boundary ecosystem has various micro-reliefs in terms of interannual variation in soil wetness conditions: always wet micro-reliefs, always dry micro-reliefs, and micro-reliefs with large interannual wetness variations. Hence, this ecosystem is a suitable area to evaluate the processes controlling $CH_4$ flux in relation to soil wetting and/or drying on an interannual timescale.

In this study, to understand relationships between $CH_4$ flux and environmental factors, we observed interannual variations in chamber $CH_4$ flux, along with the concentration, $\delta^{13}C$, and $\delta D$ of dissolved $CH_4$ during the summer, from 2009 to 2013, at the taiga-tundra boundary located on Indigirka River lowlands in northeastern Siberia. We also conducted soil incubation experiments to investigate how $\delta$ values of $CH_4$ reflect $CH_4$ production and oxidation processes in this ecosystem. In 2011, a wetting event with a significant amount of precipitation occurred. We focused in particular on the responses of $CH_4$ flux and other underlying processes to this unusual wetting event.

## 2 Methods

### 2.1 Study sites

The taiga-tundra boundary on the lowlands of the Indigirka River was selected as our study area. Observations and sampling were conducted at three sites (V: Verkhny Khatistakha, K: Kodac, and B: Boydom) in the vicinity of Chokurdakh (70° 37′ N, 147° 55′ E), Republic of Sakha (Yakutia), Russia (Fig. 1 and Table 1). The sites are located in the Russian Arctic with an annual mean air temperature of −13.9 °C and an annual mean precipitation of 208 mm for the period of 1950–2008, according to the Baseline Meteorological Data in Siberia Database (Yabuki et al., 2011). Sites V, K, and B are alongside the Indigirka River or its tributary, and tree density decreases from site V to site B.

These study sites are underlain by continuous permafrost (Iwahana et al., 2014). Normally, snowmelt and the start of active layer thawing begin in the latter half of May through the first half of June, and the growing season occurs from the end of June through the beginning of August. Air temperature and surface soil temperature (10 cm depth) peak in July, whereas the maximum thaw depth occurs from the latter half of August to the first half of September. The freezing of the active layer starts in the latter half of September to October and the whole active layer freezes from November to December.

Observations of $CH_4$ flux were conducted at seven points with three typical vegetation types, as summarized in Table 1. These vegetation types are distributed in patches, corresponding to microtopography and soil moisture (Liang et al., 2014). Micro-relief with a higher ground level is covered by green moss, larch trees, and shrubs of willows or dwarf birches. On the other hand, lower micro-relief is covered by wetland vegetation of sphagnum moss or sedges. In this study, the former vegetation type was termed 'tree mound', and the latter type was termed 'wet area'. Observation points in tree mounds were selected at each of the sites V, K, and B, and termed 'tree mound_V', 'tree mound_K', 'tree mound_B' (Table 1). For observation points of wet areas, a micro-relief covered by sphagnum moss in site K was termed 'sphagnum_K', and points covered by sedges including especially cotton-sedges (*Eriophorum angustifolium*) in sites V, K, and B were termed 'sedge_V', 'sedge_K', and 'sedge_B', respectively. Measurements of volumetric water content in the surface soil layer (0–20 cm) by TDR (time domain reflectometry; TDR-341F, Fujiwara Scientific Company, Japan) showed that tree mounds were drier than wet areas; this will be described in Sect. 3.1 (Table 1).

### 2.2 Field observations and samplings

Methane flux was observed using the chamber method in each of the typical vegetation types described in Sect. 2.1 during the summer from 2009 to 2013. A transparent cylindrical flux chamber (acrylic resin, base area $4.7 \times 10^2$ $cm^2$, height 25 cm) was installed on the ground. The headspace gas of the chamber (ca. 12 L) was circulated with a pump (ca. 1 L $min^{-1}$). The chamber was closed for 15–30 min and headspace gas was sampled two to three times after chamber closure. In most cases, the chamber was closed for 30 min and headspace gas was collected at 0 min, 15 min, and 30 min after closure. Samples were kept in pre-evacuated glass vials with butyl rubber septa. To minimize soil disturbance, we stepped on wooden boards at observation points. In 2009 and 2010, $CH_4$ flux measurements were conducted in the latter half of July, and from 2011 to

2013, observations were conducted continuously from early July to the end of July or early August. For all of these years, the observation period included the warmest season when $CH_4$ emission was expected to be the most active (Table S2).

For measurements of dissolved $CH_4$, surface water and soil pore water were sampled in wet areas from 2011 to 2013. Surface water was directly taken up by a 50 mL plastic syringe with a three-way cock attached to its tip, whereas soil pore water was sampled by a 50 mL syringe (with a three-way cock attached) through a plastic tube inserted in the soil. Soon after collecting water samples, dissolved $CH_4$ was extracted inside the syringes by the headspace method, after adding 15–35 mL of the atmosphere prepared in a 10 L aluminum bag. This atmosphere was collected beforehand at Chokurdakh village or our observation sites, and filtered by Molecular Sieves 5A (1/16 pellets, FUJIFILM Wako Pure Chemical Corporation, Japan). The atmosphere was analyzed later for $CH_4$ concentration and isotopic compositions as a background sample (2.0–4.3 ppm for $CH_4$ concentration, −53‰ to −45‰ for $\delta^{13}C$ of $CH_4$, and −168‰ to −78‰ for $\delta D$ of $CH_4$). The syringes were vigorously shaken for one minute and left standing for five minutes to ensure equilibration. Finally, headspace gas in the syringes was preserved in 10–20 mL pre-evacuated glass vials with rubber septa.

Concurrently with each flux measurement, soil temperature around the flux chamber was measured with a temperature sensor in an ORP electrode (PST-2739C, DKK-TOA Corporation, Japan) with an ORP meter (RM-30P or RM-20P). After flux measurement samples were collected, thaw depth was observed on the same day around each chamber by inserting a steel rod into the ground. From 2011 on, water level was also measured after flux measurements around each chamber in wet areas using a scale. The water level was expressed as height relative to the ground surface or the moss surface. Observation dates of these environmental factors are shown in Table S2.

**2.3 Soil incubation experiments and microbial community analysis**

Soil incubation experiments were conducted to estimate $\delta^{13}C$ and $\delta D$ of produced $CH_4$ and fractionation factors of $CH_4$ oxidation for carbon and hydrogen isotopes. For $CH_4$ production experiments, surface soil was sampled in all the wet areas in Table 1 (sedge_V, sphagnum_K, sedge_K, and sedge_B) during summer 2013. Samples were taken at 10 cm depth at each sampling location. To observe vertical variations in $\delta$ values of produced $CH_4$ within the thaw layer, we also collected samples from two additional depths (20 cm and 30 cm) at sedge_K, which is a location typical of the taiga-tundra boundary region. These samples were from organic layers, except for the samples from 30 cm, which were from the top of the mineral layer.

Approximately 10 mL of soil was directly transferred into each plastic syringe (60 mL maximum capacity) along with in situ water (approximately 50 mL) to prevent the sample from being oxidized by the atmosphere. Syringes were preserved in water to ensure no leakage and were immediately pre-incubated for 4–8 days, then incubated in triplicate for 8 days. Pre-incubation and incubation temperatures were set at 5 °C. We also incubated syringes at 10 °C for samples from 10 cm depth at sedge_K to investigate temperature dependence of $\delta$ values of produced $CH_4$. For each of these seven incubation treatments (sphagnum_K, sedge_V, sedge_K, and sphagnum_K, 10 cm depth, 5 °C; sedge_K, 20 cm and 30 cm depths, 5 °C; sedge_K, 10 cm depth, 10 °C), three replicate soil samples were prepared. Water in each incubation syringe was

sampled twice at the start and the end of incubation, and dissolved $CH_4$ was extracted using the headspace method described in Sect. 2.2. As a consequence, dissolved $CH_4$ samples were collected in triplicate for each of the initial and final conditions of one incubation treatment.

To interpret $CH_4$ production in the incubation experiments (Sect. 2.3), phylogenic composition of methanogens in the surface soil was additionally analyzed in 2016 using 16S rRNA gene sequencing. In July 2016, soil samples from 10 cm depth were collected in 10 mL plastic tubes in triplicate in the same four wet areas as the anaerobic incubation experiments, and kept frozen until analysis. DNA was extracted from 3 g of the soil samples as described by Ikeda et al. (2004). Extracted DNA was purified using the OneStep™ PCR Inhibitor Removal Kit (Zymo Research, Calif.) and quantified using the Quant-iT PicoGreen dsDNA assay Kit (Invitrogen, Carlsbad, Calif.). Amplicon sequencing was conducted targeting the V3/V4 regions of 16S rRNA genes (Caporaso et al. 2011). Sequences obtained were processed through the QIIME pipeline (Caporaso et al. 2010). A representative sequence was picked from each operational taxonomic unit (OTU), and the Greengenes reference database (version 13.8) was used to assign taxonomic information and calculate the relative abundance of methanogenic archaea present.

For $CH_4$ oxidation, surface organic layers (0‑13 cm depth) were sampled at sphagnum_K and sedge_K in July 2012, and then kept in a refrigerator until the experiment (6 days). These soil samples were cut into small pieces and mixed well with air. Ten grams (about 40 mL) of soil sample were transferred into plastic syringes (maximum 120 mL) in quadruplicate for each sampling location. Approximately 80 mL of air and 0.2–2 mL of 25% $CH_4$ gas were added to each syringe so the total volume in each syringe was 120 mL and the headspace $CH_4$ concentration was $5.0 \times 10^2$–$4.8 \times 10^3$ ppm. Syringes were preserved in water and incubated at 8 °C for 8 days. Headspace gas was sampled on day 0, day 4, and day 8 from each syringe into 20 mL pre-evacuated glass vials with rubber septa. Consequently, quadruplicate gas samples were collected for each location and each sampling day.

## 2.4 Sample analysis and data processing

Methane concentrations in air samples were analyzed using a gas chromatograph (HP6890 series G1530A, Hewlett Packard, USA) equipped with a flame ionization detector and a CP-carboplot capillary column (Varian, USA). Methane flux was calculated from $CH_4$ concentration in chamber headspace by a linear regression of two to three concentration values against the time elapsed since chamber closure. The detection limit of $CH_4$ flux for each observation was calculated as 0.8–2.4 mg $CH_4$ $m^{-2}$ $day^{-1}$, based on whether the change of chamber $CH_4$ concentration during the observation was significant relative to the precision of $CH_4$ concentration analysis. Regression $r^2$ was calculated (formally) as $\geq 0.87$, when the flux value was larger than 2 mg $CH_4$ $m^{-2}$ $day^{-1}$. Dissolved $CH_4$ concentrations were obtained from calculation of the headspace method where equilibrations of $CH_4$ between gas and water phases are described by the Bunsen absorption coefficient of $CH_4$ (Yamamoto et al., 1976).

Carbon and hydrogen isotope ratios of in situ dissolved $CH_4$ and $CH_4$ samples from both incubation experiments were analyzed on a GC/GC/C/IRMS (modified after Sugimoto, 1996) —which is a continuous flow system consisting of two

gas chromatographs, a combustion reactor, and an isotope ratio mass spectrometer (MAT253, Thermo Fisher Scientific, USA) —and on a GC/GC/P/IRMS (P: pyrolysis in a HTC reactor of GC IsoLink, Thermo Fisher Scientific), respectively. Carbon and hydrogen isotope ratios obtained were represented relative to VPDB and VSMOW, respectively. Precisions of the analyses were $\pm 0.2‰$ and $\pm 2‰$ for $\delta^{13}C$ and $\delta D$, respectively. When calculating $\delta^{13}C$ and $\delta D$ of dissolved $CH_4$, the effect of $CH_4$ in background air was removed based on the mass balance. In the aerobic incubation experiments, the fractionation factors of $CH_4$ oxidation for carbon and hydrogen were calculated using the following Rayleigh distillation equation:

$$\ln \frac{R_t}{R_0} = \left( \frac{1}{\alpha_{ox}} - 1 \right) \ln \frac{[CH_4]_t}{[CH_4]_0}, \tag{1}$$

where $R_0$ and $R_t$ represent isotope ratios under initial conditions and at time $t$, respectively; $\alpha_{ox}$ is the fractionation factor for $CH_4$ oxidation (defined so that $\alpha_{ox} > 1$); and $[CH_4]_0$ and $[CH_4]_t$ are $CH_4$ concentrations under initial conditions and at time $t$, respectively.

All statistical tests for detecting differences in $CH_4$ fluxes or dissolved $CH_4$ concentrations were conducted using R software (version 3.3.3). Mann-Whitney's U test was applied to compare magnitudes between two years of data and Steel-Dwass's multiple comparison test was used to compare magnitudes among three years or more of data.

## 2.5 Meteorological data

Air temperature and precipitation observed at a weather station in Chokurdakh (WMO station 21946) were used to investigate interannual variations in meteorological conditions during our observation period of $CH_4$ flux (2009‑2013) and during the preceding two years (2007–2008). The distance between the weather station and our farthest observation site (site V) is approximately 45 km (Fig. 1). These data were obtained from GHCN-Daily, a NOAA database (Menne et al., 2012a, 2012b).

## 3 Results

### 3.1 Environmental factors

Soil wetness conditions and thaw depth differed among vegetation types (Table 1). Tree mounds had lower surface water content (2.1–17%) than wet areas (42–48%). Among the two types of wet areas, the water level was lower in wet areas of sphagnum mosses than those of sedges (Fig. 2). Wet areas of sedges experienced water levels higher than the ground surface (defined as 0 cm), reaching more than 10 cm above the ground surface. Corresponding with soil water content, the thaw depth was shallower at dry tree mounds (20–23 cm), and deeper in wet areas (31–56 cm). In wet areas, thaw depth became deeper from 2011 ($22 \pm 4$ cm) to 2012 ($25 \pm 8$ cm) and 2013 ($35 \pm 7$ cm) in observations made during mid-July (Table S1). The overall average thaw depth observed on days when flux measurements were taken was $31 \pm 12$ cm ($n = 77$, 9–58 cm between Jul 3 and Aug 9).

Figure 2 shows persistently low annual precipitation (162–173 mm) from 2007 to 2009. In 2010, July air temperature was characteristically high (15.5 °C) accompanying low monthly precipitation (8 mm). These show dry conditions during our flux observations in 2009 and 2010. Parmentier et al. (2011) reported that water level was lower in summer 2009 than the previous two summers at a tundra research station (Kytalyk) in the vicinity, approximately 30 km to northwest of Chokurdakh. In contrast, precipitation in July 2011 was extremely high (94 mm) with a relatively mild temperature (13.0 °C), which caused an unusual wetting. High precipitation continued in August (74 mm) and September (67 mm) of the same year. Corresponding with this heavy rainfall, water levels were also high in 2011, and subsequent observations show a clear decrease from 2011 to 2013 in wet areas of sedges ($p < 0.005$). Water levels also declined in wet area of sphagnum mosses, with values of −9 cm, −10 cm, and less than −12 cm in 2011, 2012, and 2013, respectively.

## 3.2 $CH_4$ flux and dissolved $CH_4$ concentration

Obtained $CH_4$ flux shows clear spatial and interannual variations (Fig. 3). Individual flux measurements ($n = 143$ in total) are summarized as mean values for the main summer seasons. From 2011 to 2013, continuous flux observations (Table S2) were conducted in concert with dissolved $CH_4$ analysis, and the interannual variation during this period will be discussed in detail.

With regards to the spatial variation of $CH_4$ flux, tree mounds had consistently small values around the detection limit for all measurements (−4.9 to 1.9 mg $CH_4$ $m^{-2}$ $day^{-1}$), while wet areas showed $CH_4$ emissions. From 2009 to 2013, the $CH_4$ flux in wet areas showed large interannual variations ranging from 36 to 140 mg $CH_4$ $m^{-2}$ $day^{-1}$. The flux increased in 2011 when the wetting event occurred, then remained relatively large in 2012 (compared to 2009 and 2010). Moreover, the flux increased again from 2011/2012 to 2013 ($p < 0.05$). No statistically significant correlation was found when $CH_4$ flux was plotted against soil temperature (10 cm depth), thaw depth, or water level using all the data from wet areas (Fig. S1).

In addition to $CH_4$ flux, dissolved $CH_4$ concentration increased after the wetting event in 2011 (Fig. 4). From 2011 to 2012, $CH_4$ concentration in soil pore water at 10 cm depth (Fig. 4b) exhibited a sharp increase of one order of magnitude ($p < 0.005$). It remained high from 2012 to 2013, and the concentrations in surface water and that at 20 cm depth (Fig. 4a and c) also increased significantly over the same period ($p < 0.05$). No significant difference in concentration was observed at 30 cm depth between 2012 and 2013. In terms of vertical variation, the concentration in surface water was lower than that in soil pore water (10, 20, and 30 cm depth).

## 3.3 $\delta^{13}C$ and $\delta D$ of in situ dissolved $CH_4$

Variability of both $\delta^{13}C$ and $\delta D$ of dissolved $CH_4$ was smaller in deeper layers, showing different patterns between $\delta^{13}C$ and $\delta D$, and across years (Fig. 5). The $\delta^{13}C$ of dissolved $CH_4$ had similarly large ranges (−68‰ to −40‰) in surface water and at 10 cm and 20 cm depths, compared to a small range (−53‰ to −46‰) at 30 cm depth. The $\delta D$ of dissolved $CH_4$ was variable only in surface water (−415‰ to −308‰) and at 10 cm depth (−417‰ to −341‰), whereas it had a constant value of around −408‰ at 20 cm and 30 cm depths. Additionally, $\delta^{13}C$ values approached a relatively high value (approximately −50‰) at depth, while $\delta D$ values converged to almost their lowest value. In terms of interannual variations in $\delta^{13}C$ and $\delta D$ of dissolved

CH$_4$ from 2011 to 2013, both δ$^{13}$C and δD values in surface soil pore water (10 cm depth) were scattered more widely in 2011, showing standard deviations (SD) of 6.6‰ and 24‰, respectively, whereas their ranges were smaller in 2012 and 2013 (SD: 3.3‰ and 17‰ at maxima, respectively).

As shown in Fig. 6, convergence of δ$^{13}$C and δD of dissolved CH$_4$ is associated with dissolved CH$_4$ concentrations. The δ$^{13}$C and δD values of dissolved CH$_4$, including surface water and 10 cm depth, converged at high CH$_4$ concentrations to the values seen in deeper soil layers (δ$^{13}$C = −50 ± 5‰ and δD = −408 ± 5‰ at > 200 μmol CH$_4$ L$^{-1}$).

### 3.4 Soil incubation experiments and microbial community analysis

In the anaerobic incubation experiment, the CH$_4$ production rate was different among sampling locations (Fig. 7); the rate was higher for sedge_K and sedge_B (0.66 ± 0.15 μmol day$^{-1}$ and 0.43 ± 0.09 μmol day$^{-1}$, respectively) than sedge_V and sphagnum_K (0.24 ± 0.02 μmol day$^{-1}$ and 0.08 ± 0.01 μmol day$^{-1}$, respectively). In sedge_K, the sampling location tested in detail, production was more rapid for shallower soil layers among the 10 cm, 20 cm and 30 cm depths (0.66 ± 0.15 μmol day$^{-1}$, 0.33 ± 0.06 μmol day$^{-1}$, 0.003 ± 0.004 μmol day$^{-1}$, respectively; $p < 0.01$ in Welch's ANOVA test), while no difference in the rate was found between incubation temperatures (0.66 ± 0.15 μmol day$^{-1}$ gdw$^{-1}$ at 5 °C and 0.74 ± 0.14 μmol day$^{-1}$ gdw$^{-1}$ at 10 °C, $p > 0.5$ in $t$-test). When the CH$_4$ production rate was high, the δ$^{13}$C and δD values of produced CH$_4$ were less variable irrespective of sampling location, sampling depth, or incubation temperature. The δ$^{13}$C value of produced CH$_4$ at a high production rate (> 0.26 μmol day$^{-1}$ gdw$^{-1}$) was −55 ± 4‰ (n = 12). Similarly, δD under rapid CH$_4$ production was −410 ± 9‰ (n = 12). These δ$^{13}$C and δD values of CH$_4$ obtained under rapid production were mostly comparable with the δ values of in situ dissolved CH$_4$ that converged in deep soil layers (δ$^{13}$C = −50 ± 2‰ at 30 cm depth and δD = −408 ± 5‰ at 20–30 cm depth; Fig. 5c and d), although δ$^{13}$C values in the incubation experiment were slightly lower than those in situ.

In the microbial community analysis using 16S rRNA gene sequencing (Fig. 8), soil with high rates of CH$_4$ production shown in the incubation experiment (sedge_K and sedge_B as in Fig. 7) had more abundant total methanogens within the detected archaea than that with slow CH$_4$ production rates (sphagnum_K and sedge_V). Acetoclastic methanogens in the order Methanosarcinales were higher in proportion among methanogens in sedge_K and sedge_B, where δ$^{13}$C values of produced CH$_4$ in the incubation were higher. In contrast, Methanosarcinales were fewer in proportion in sphagnum_K, where δ$^{13}$C of the produced CH$_4$ was lower.

In the CH$_4$ oxidation experiment, CH$_4$ concentration in headspace declined continuously in every sample (Fig. S2). As CH$_4$ oxidation proceeded, both δD and δ$^{13}$C of the remaining CH$_4$ increased with a linear relationship between them (Fig. 9, S2). Observed slope Δ(δD)/Δ(δ$^{13}$C) was 11, indicating a much larger fractionation of hydrogen than carbon, regardless of vegetation types in wet areas (sphagnum or sedge). The hydrogen isotope fractionation factors of CH$_4$ oxidation calculated from the data shown in Fig. 9 were 1.25 and 1.16 for wet areas of sphagnum and sedge, respectively, while carbon isotope fractionations were 1.021 and 1.015, respectively.

# 4 Discussion

## 4.1 CH$_4$ flux at tree mounds and wet areas at the taiga-tundra boundary on the Indigirka River lowlands

Methane flux observed in our study was clearly larger at wet areas than at dry tree mounds (Table 1, Fig. 3). Such differences in CH$_4$ flux between wetland vegetation and dry areas with trees or shrubs is generally observed (van Huissteden et al., 2005; van der Molen et al., 2007; Flessa et al., 2008) and is consistent with the fact that CH$_4$ production requires reducing conditions in soil (Conrad, 2007). Our CH$_4$ flux in wet areas (36–140 mg CH$_4$ m$^{-2}$ day$^{-1}$) was comparable to that reported for wet tundras (32–101 mg CH$_4$ m$^{-2}$ day$^{-1}$) or permafrost fens (42–147 mg CH$_4$ m$^{-2}$ day$^{-1}$) in a database across permafrost zones complied by Olefeldt et al. (2013). In forests, many studies have observed CH$_4$ absorption instead of emission (King et al., 1997; Dutaur and Verchot, 2007; Flessa et al., 2008; Morishita et al., 2014). However, our observations at tree mounds rarely found CH$_4$ absorption or emission. In addition, CH$_4$ was not consumed even under O$_2$- and CH$_4$-rich conditions in incubation experiments of tree mound soil from site K (Murase et al., 2014), indicating that a lack of methanotrophic bacterial activity limited CH$_4$ absorption at this vegetation type. Our results show that CH$_4$ emission from wet areas is expected to make a greater contribution to ecosystem-scale CH$_4$ exchange at the taiga-tundra boundary on the Indigirka River lowlands.

## 4.2 Methane flux, production, and oxidation responses to the wetting event

In 2009 and 2010 the CH$_4$ emission in wet areas was low (Fig. 3), even at relatively high soil temperatures in 2010 (Fig. S1), under dry conditions that were not directly observed in this study. The wetting event in 2011 initiated the high CH$_4$ emission that continued through 2013 despite decreasing water levels (Fig. 2). Moreover, a further increase in flux was observed in 2013, accompanying a build-up of dissolved CH$_4$ (2011– 2013) as shown in Fig. 4.

These interannual variations from 2011 to 2013 could be caused by the development of reducing soil conditions over multiple years after the wetting event. Reducing soil conditions may have developed, to some extent, as a result of the extreme precipitation in the summer of 2011 (Fig. 2). The surface soil layer, particularly under high water levels, could eliminate O$_2$ from soil pore spaces due to water saturation. These reducing conditions may have been preserved by freezing of the soil throughout the following winter. Additionally, a surface soil layer saturated with ice could have prevented snowmelt water (rich in O$_2$) from infiltrating the soil during the spring thaw season of 2012 (Woo, 2012). These processes would have led to the continuation of reducing conditions in the soil, which were created in summer 2011, into 2012. Through further decomposition of soil organic matter with the consumption of O$_2$, reducing soil conditions may have been exacerbated in the water-saturated soil layer to a greater extent in summer 2012. Water in the saturated soil layer may be retained without exchange for a relatively long time in our study sites, because lateral runoff is assumed to be small in the flat floodplain that comprises the Indigirka River lowlands (Nassif and Wilson, 1975). In addition, deep percolation loss is prevented by the impermeable permafrost layer (Woo, 2012). From summer 2012 to summer 2013, reducing conditions in the soil may have been similarly prolonged, especially in the deep soil layer, despite the decrease in water level from

summer 2011 to summer 2013. This continuous soil reduction from 2011 to 2013 could have promoted $CH_4$ production and/or decreased $CH_4$ oxidation, which may explain the increase in dissolved $CH_4$ concentration and $CH_4$ flux in wet areas following the wetting event and continuing until 2013 (Fig. 3 and 4).

In 2011, $\delta^{13}C$ and $\delta D$ of dissolved $CH_4$ (10 cm depth) were scattered broadly across a wide range, whereas in 2012 and 2013 the ranges were narrower and they clustered around a high $\delta^{13}C$ value (−50‰) and a low $\delta D$ value (−408‰; Fig. 5b). Considering that $\delta D$ increased much more rapidly than $\delta^{13}C$ in our oxidation experiment (Fig. 9), $\delta D$ can be considered as a sensitive indicator of $CH_4$ oxidation. In contrast, $\delta^{13}C$ is not a good indicator because its fractionation factor of $CH_4$ oxidation (1.015‑1.021) was similar to that of $CH_4$ diffusion (1.019; Chanton, 2005), thus the effects of $CH_4$ oxidation and diffusion cannot be discerned by $\delta^{13}C$. Additionally, $\delta D$ of dissolved $CH_4$ (Fig. 5) was clearly lower in deeper layers (20 cm and 30 cm depths) than in shallower layers (surface water and 10 cm depth), which indicates $\delta D$ showed $CH_4$ oxidation in situ as well, because shallower layers are provided with $O_2$ from the atmosphere and precipitation. The effect of $CH_4$ exchange between surface dissolved $CH_4$ and atmospheric $CH_4$ can be excluded, because all the dissolved $CH_4$ observed in this study was highly oversaturated (> 0.3 μmol $L^{-1}$, Fig. 4) compared to the equilibrium concentration of atmospheric $CH_4$ (4–5 nmol $L^{-1}$, assuming 1–10 °C water temperature and 2 ppm atmospheric $CH_4$ concentration; Yamamoto et al., 1976). Thus, $\delta D$ values at 10 cm in 2011 were scattered broadly compared with those in 2012 and 2013 that clustered around a low value, suggesting that $CH_4$ oxidation was significant in the surface soil layer during the year of the wetting event (2011). In 2012 and 2013, $CH_4$ oxidation became insignificant, relative to the larger pool of dissolved $CH_4$ (Fig. 4). In the $CH_4$ production incubation experiment, $\delta^{13}C$ and $\delta D$ of produced $CH_4$ were less variable at higher production rates ($\delta^{13}C = −55 \pm 4$‰ and $\delta D = −410 \pm 9$‰ as in Fig. 7). Analogously, those of in situ dissolved $CH_4$ converged at a high $CH_4$ concentration around similar values ($\delta^{13}C = −50 \pm 5$‰ and $\delta D = −408 \pm 5$‰ in Fig. 6). This suggests that $\delta$ values of produced $CH_4$ became almost constant under rapid $CH_4$ production in situ and that the convergence of $\delta$ values of dissolved $CH_4$ observed in situ reflect rapid $CH_4$ production. Hence, the narrow ranges of $\delta^{13}C$ and $\delta D$ values of dissolved $CH_4$ at 10 cm depth observed in 2012 and 2013 ($\delta^{13}C$: around −50‰ and $\delta D$: around −408‰, Fig. 5b) suggest enhanced $CH_4$ production relative to the wetting year (2011).

Multi-year effects of wetting on $CH_4$ flux through soil reduction processes have been previously proposed by Kumagai and Konno (1998) and Desyatkin et al. (2014) as one possible factor for explaining the increase in $CH_4$ flux after wetting. Kumagai and Konno (1998) reported a $CH_4$ flux increase at a temperate rice field in Japan one year after the rice field was irrigated and restored from farmland that had been drained for eight years. Desyatkin et al. (2014) observed flux increases at a thermokarst depression in boreal eastern Siberia during the second consecutive year of flooding following large volumes of precipitation. On the other hand, studies at natural wetlands in the northeastern USA (Smemo and Yavitt, 2006; Treat et al., 2007; Olson et al., 2013) and southern Canada (Moore et al., 2011) reported that interannual variations of $CH_4$ flux correspond with those of water level and/or precipitation in the current year. In our study area, multi-year soil reduction may be important because soil temperature is generally lower than 11 °C (10 cm to 30 cm depth; Fig. S1a and Iwahana et al., 2014) due to a shallow active layer underlain by permafrost. Therefore, decomposition of organic matter can

be slow (Treat et al., 2015), which would slowly decrease soil redox potential, allowing it to remain relatively high in the first year of wetting.

In the following two years (2012–2013), we observed redox potential values lower than −100 mV in wet areas (Table S6), which are well below the upper limit for $CH_4$ production in soil (Conrad, 2007; Street et al., 2016). Methane production at a potential higher than −100 mV can also occur, because soil is heterogeneous and can have more reducing microsites than the rest of the bulk soil, where redox potential can be measured (Teh et al., 2005; Teh and Silver, 2006).

In addition to the multi-year soil reduction, it appears that the wetting event led to the thaw depth increase in wet areas from 2011 to 2013 (Table S1). Although thaw depth increased, summer air temperatures decreased from 2011 (7.7, 13.0 °C as June and July mean temperatures, respectively) to 2012 (7.4, 9.2 °C) and 2013 (6.6, 10.5 °C) as shown in Fig. 2. The wetting event may have led to the $CH_4$ flux increase from 2011 to 2013 (Fig. 3) partly through the thaw depth increase, by thickening the soil layer where $CH_4$ production occurs (Nakano et al., 2000; van Huissteden et al., 2005). However, the clear increase in dissolved $CH_4$ concentration (Fig. 4) and the enhanced $CH_4$ production and less significant $CH_4$ oxidation reflected in our isotopic data (Fig. 5b) cannot be explained by the thaw depth increase. Additionally, in the incubation experiment of $CH_4$ production (Fig. 7), the $CH_4$ production rate under anaerobic conditions was slower in the deeper layer, especially at 30 cm depth (mineral soil) compared to 10 cm and 20 cm depths (organic soil) in sedge_K. Treat et al. (2015) also reported, from a pan-Arctic synthesis of anaerobic incubations, that difference in soil types (organic/mineral) and that in substrate quality along depth are important controls on $CH_4$ production rate. Our results from the incubations suggests that the deep layer comprised of mineral soil, where $CH_4$ production becomes active when thaw depth increase, is not the main region for $CH_4$ production.

This study did not evaluate vegetation cover quantitatively, and the wetting event might have also led to some vegetation change (such as increase of sedges), although no drastic changes were found visually in the observed wet areas. Increase in cover by sedges might have raised $CH_4$ flux partly by providing labile organic substrate for $CH_4$ production or conduits for the $CH_4$ transport from the soil to the atmosphere (Chanton, 2005; Lai, 2009; Ström et al., 2015).

## 4.3 Process behind $CH_4$ production response

When $CH_4$ production is initiated after the onset of anoxia in rice paddy soil, it first occurs via hydrogenotrophic methanogenesis, and then by both hydrogenotrophic and acetoclastic methanogenesis, which increases $CH_4$ production rate (Conrad, 2007). Afterwards, the ratio of acetoclastic to hydrogenotrophic methanogenesis can stabilize (Roy et al., 1997). Considering that this ratio is an important control on isotopic compositions of produced $CH_4$, stabilization of production pathways might explain the convergence in δ values of dissolved $CH_4$ at our study sites under high $CH_4$ concentration (Fig. 6), and the reduced variability of δ values of produced $CH_4$ in our experiment under rapid production conditions (Fig. 7). As acetoclastic methanogenesis leads to higher $\delta^{13}C$ in produced $CH_4$ than hydrogenotrophic methanogenesis (Sugimoto and Wada, 1993), acetoclastic methanogenesis may have been activated when dissolved $CH_4$ concentration or $CH_4$ production rate were high during our study. This interpretation is supported by the microbial community analysis (Fig. 8), where

acetoclastic methanogens of Methanosarcinales were more abundant in wet areas, with a higher $\delta^{13}C$ of produced $CH_4$ in the incubation. Therefore, the high and less-variable $\delta^{13}C$ values observed at 10 cm depth in 2012 and 2013 (Fig. 5b) suggest a greater contribution from acetoclastic methanogenesis compared to the wetting year (2011). Similar to findings from rice paddy soil (Conrad, 2007), acetoclastic methanogenesis may have experienced delayed activation after anoxic conditions

began in 2011, which could also have promoted $CH_4$ production in 2012 and 2013.

## 5 Concluding remarks

At the taiga-tundra boundary on the Indigirka River lowlands, we observed an increase in $CH_4$ flux in wet areas following the wetting event in 2011, and a further increase in flux in 2013. Our results show interannual variations in $\delta^{13}C$ and $\delta D$ of dissolved $CH_4$, and when compared with our incubation experiments, suggest both enhancement of $CH_4$ production and less

significance of $CH_4$ oxidation in 2012 and 2013 compared to 2011. This enhancement of production could be partly caused by activation of acetoclastic methanogenesis following the development of reducing soil conditions after the wetting event. Analyses of isotopic compositions of $CH_4$ both in situ and in incubation experiments can be combined to investigate the effects of $CH_4$ production and oxidation on these isotopic compositions, and to clarify the relationship between $CH_4$ flux and wetting. In the future, measuring the $\delta^{13}C$ of dissolved $CO_2$ would be useful to further validate activation of acetoclastic

methanogenesis (Sugimoto and Wada, 1993; McCalley et al., 2014; Itoh et al., 2015). Outside of these processes, the wetting event might have affected $CH_4$ flux partly via the thaw depth increase or some amount of vegetation change. It would be useful to analyze $\delta^{13}C$ and $\delta D$ values of emitted $CH_4$ in order to assess changes in $CH_4$ transport (such as by increase of sedge cover) and to investigate the relationship between dissolved $CH_4$ concentration and $CH_4$ flux in detail (Chanton, 2005).

In recent years, strong storm activity and wetting events in terrestrial ecosystems have been observed in northern

regions (Iijima et al., 2016). A wetting event at the taiga-tundra boundary can switch micro-reliefs with large interannual variations in soil wetness conditions to significant $CH_4$ sources; we observed clear increases in $CH_4$ flux at wet areas after the wetting event. In order to predict $CH_4$ flux following a wetting event in a permafrost ecosystem, our results show the multi-year process of soil reduction affected by the duration of water saturation in the active layer.

**Author contribution**

Ryo Shingubara and Atsuko Sugimoto designed the experiments and Ryo Shingubara carried them out. Go Iwahana, Shunsuke Tei, Liang Maochang, Shinya Takano, Tomoki Morozumi, and Trofim C. Maximov helped with sampling, in situ data collection, and preparing resources for the fieldwork. Jun Murase contributed to the laboratory analysis. Ryo Shingubara prepared the manuscript with contributions from all co-authors.

**Competing interests**

The authors declare that they have no conflict of interest.

**Acknowledgments**

This research was supported by JSPS Grant-in-Aid for Scientific Research (KAKENHI) numbers 21403011 and 16H05618, a Grant-in-Aid from the Global COE Program 'Establishment of Center for Integrated Field Environmental Science' (IFES-GCOE) funded by the Ministry of Education, Culture, Sports, Science and Technology-Japan (MEXT), JST (Strategic International Collaborative Research Program: SICORP)-EU cooperative research project 'Dynamics of permafrost and methane emission in Arctic terrestrial ecosystem in Eastern Siberia', the Green Network of Excellence (GRENE) Program funded by MEXT, and the COPERA (C budget of ecosystems and cities and villages on permafrost in eastern Russian Arctic) project funded by the Belmont Forum through JST. We sincerely thank A. Kononov, R. Petrov, E. Starostin, A. Alexeeva and other members of the Institute for Biological Problems of Cryolithozone SB RAS, and T. Stryukova, S. Ianygin and other staff at the Allikhovsky Ulus Inspectorate of Nature Protection for supporting our fieldwork in the vicinity of Chokurdakh. We also wish to acknowledge help from Y. Hoshino, S. Nunohashi, K. Tanaka, K. Saito, H. Kudo, and R. Shakhmatov in our research group at Hokkaido University.

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

**Table 1. Observation points of chamber CH$_4$ flux. Concentration and isotopic compositions of dissolved CH$_4$ were also observed in the following wet areas.**

| Site | Landscape | Observation points and surface conditions | Dominant vegetation | Volumetric water content (%)[b] | Thaw depth (cm)[c] |
|---|---|---|---|---|---|
| V (Verkhny Khatistakha) 70° 15' N 147° 28' E | Larch forest and wetland | tree mound_V | Green moss, *Larix gmelinii* | 17 ± 5 (*n* = 3) | 23 ± 3 (*n* = 5) |
| | | sedge_V (wet area) | *Carex spp.*, *Eriophorum angustifolium* | 48 ± 4 (*n* = 3) | 56 ± 3 (*n* = 4) |
| K (Kodac)[a] 70° 34' N 148° 16' E | Typical taiga-tundra boundary | tree mound_K | Green moss, *Larix gmelinii* | 2.1 ± 0.6 (*n* = 4) | 23 ± 4 (*n* = 9) |
| | | sphagnum_K (wet area) | *Sphagnum squarrosum* | 42 ± 5 (*n* = 6) | 31 ± 8 (*n* = 15) |
| | | sedge_K (wet area) | *Eriophorum angustifolium* | 44 ± 4 (*n* = 6) | 32 ± 13 (*n* = 28) |
| B (Boydom) 70° 38' N 148° 09' E | Low-centered polygon | tree mound_B | Green moss, *Larix gmelinii* | 6 ± 2 (*n* = 5) | 20 ± 4 (*n* = 8) |
| | | sedge_B (wet area) | *Eriophorum angustifolium* | 46 ± 2 (*n* = 5) | 36 ± 9 (*n* = 8) |

[a] Site K was previously named as Kryvaya (Iwahana et al., 2014) or Kodak (Liang et al., 2014).

[b] Observed for the surface soil layer down to 20 cm on 1 to 3 days in July 2011 at each observation point (see Table S2 for detailed observation dates). Standard deviations are shown.

[c] Observed from early July to early August during 2010–2013 (see Table S1 for the interannual variation and Table S2 for detailed observation dates). Standard deviations are shown.

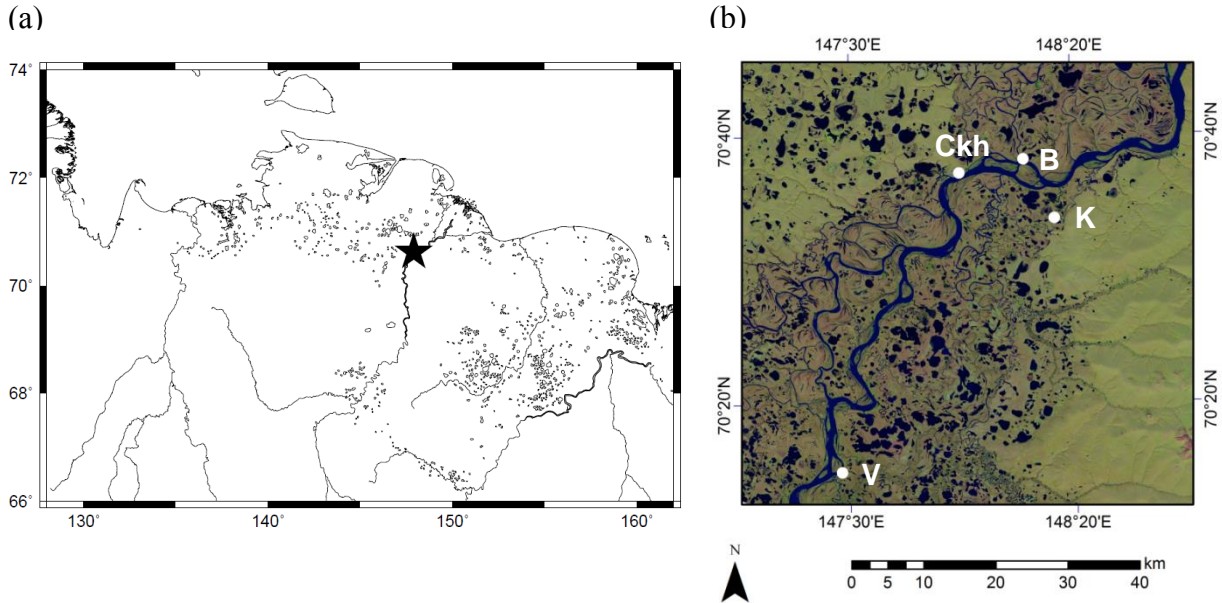

Figure 1: Locations of study sites. (a) Location of study region in northeastern Siberia (the Generic Mapping Tools 5.0.0). (b) Satellite image of Indigirka River lowland around Chokurdakh village (Ckh: 70° 37' N, 147° 55' E) from Landsat 8. Observation sites (V, K, B) were selected in this region alongside the main stem and a tributary of the Indigirka River.

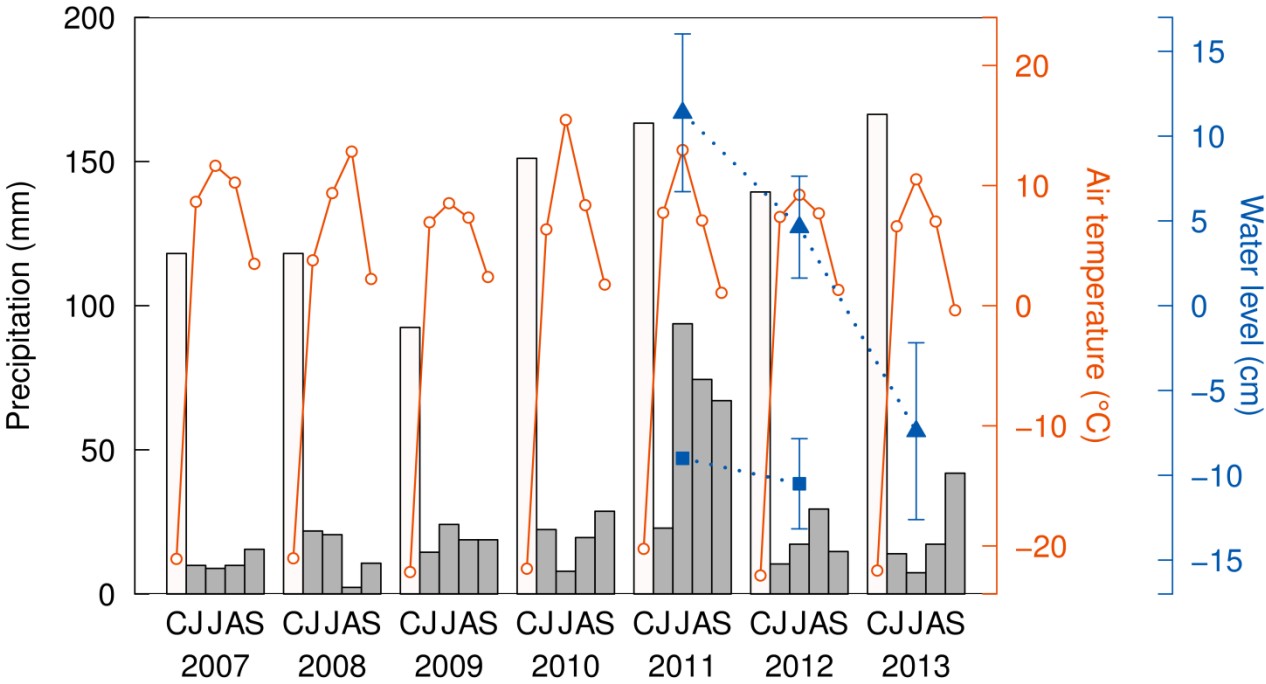

**Figure 2: Interannual variations in precipitation (bars) and air temperature (solid lines) observed at a weather station at Chokurdakh for the cold season with snow cover (C: total from October in the previous year to May in the current year) and the warm season (JJAS), and water level (dotted lines) measured in wet areas of sedges (triangle) and sphagnum mosses (square). Water level was very low (< −12 cm) in the wet area of sphagnum in 2013, and could not be measured. Error bars represent standard deviations. Methane flux was observed during the main summers (early July to early August) from 2009 to 2013.**

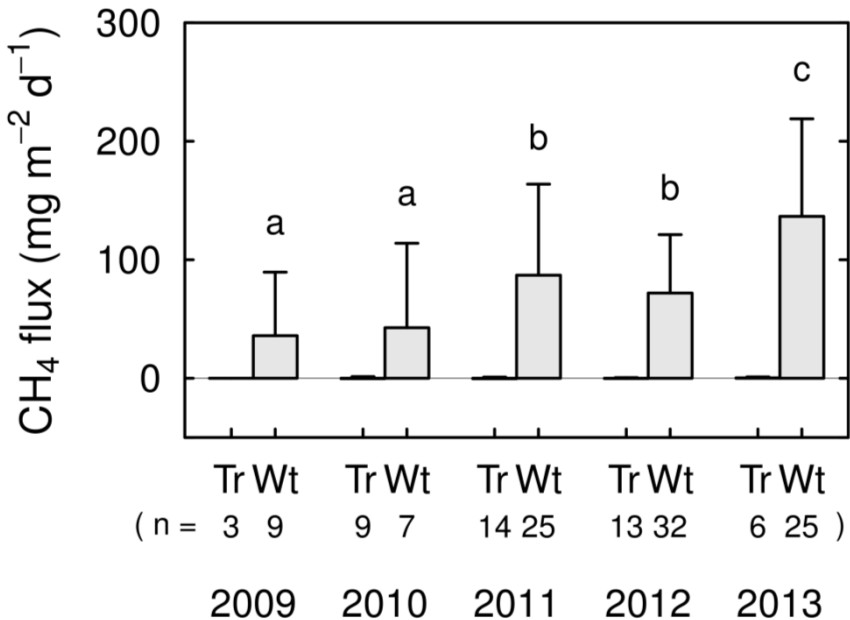

**Figure 3: Interannual variations in averaged CH$_4$ flux in tree mounds (denoted as "Tr") and wet areas ("Wt") for main summer seasons from 2009 to 2013. Replication numbers ("n") are shown for each averaged flux value, and standard deviations are represented by error bars. Different letters show statistical interannual differences in the flux values for wet areas. See Table S2 for flux values at respective observation points.**

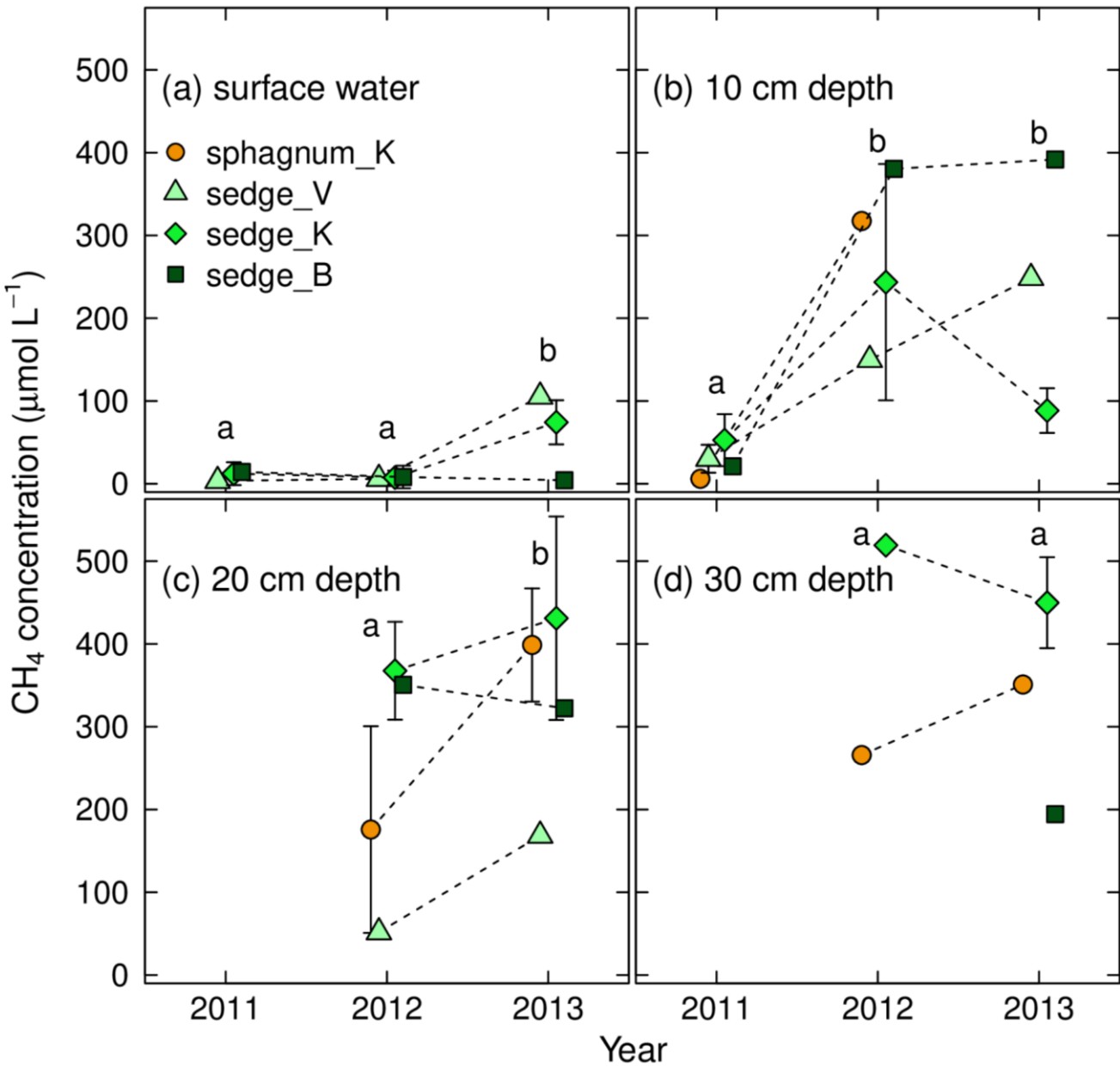

**Figure 4: Interannual variation in dissolved CH$_4$ concentration in (a) surface water and soil pore water at (b) 10 cm, (c) 20 cm, and (d) 30 cm depths in wet areas from 2011 (wetting event) to 2013. Different letters in each panel denote statistical differences among years in averaged concentration across the four wet areas ($p < 0.05$). Error bars represent standard deviations. See Table S3 for numerical values of dissolved CH$_4$ concentrations.**

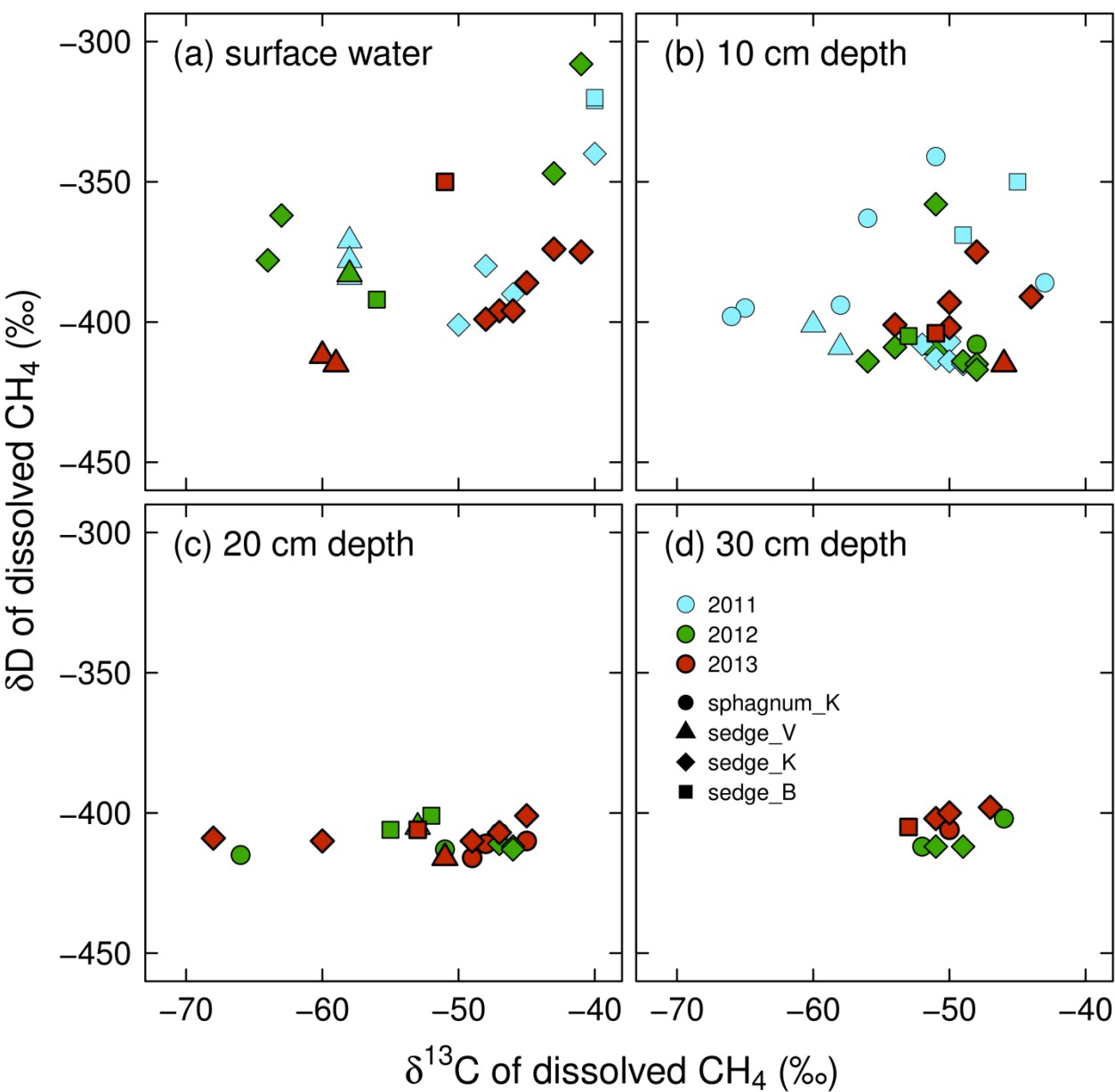

Figure 5: In situ $\delta^{13}C$ versus $\delta D$ of dissolved $CH_4$ in (a) surface water and soil pore water at (b) 10 cm depth, (c) 20 cm depth, and (d) 30 cm depth from the wet event in 2011 to 2013. Individual delta values are shown here. See Table S3 for averaged delta values for each observation point and each year.

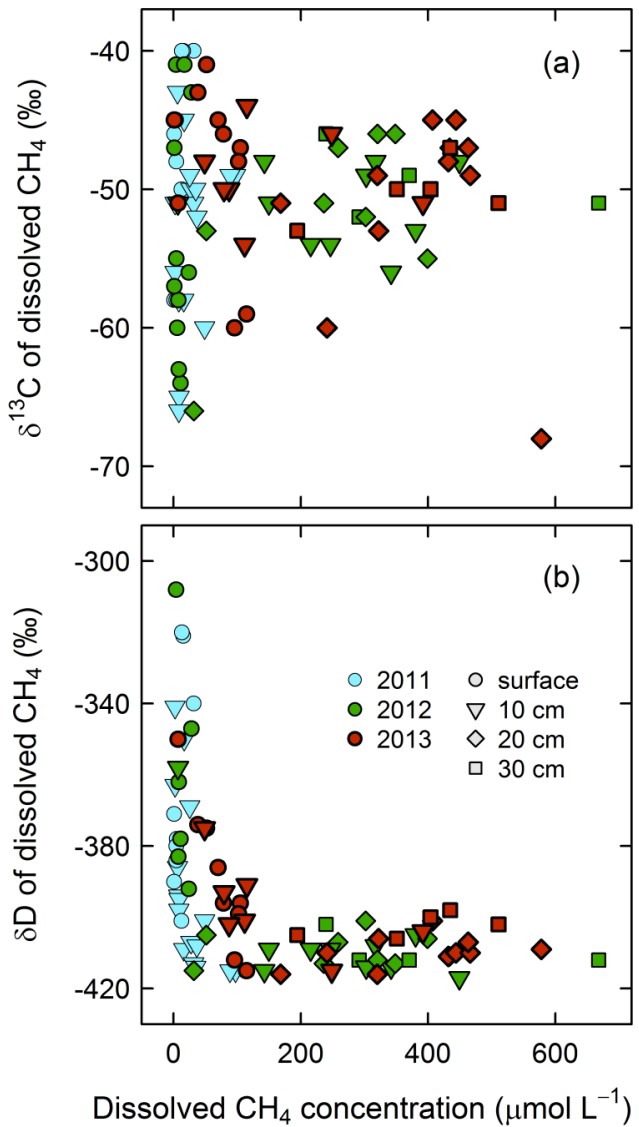

**Figure 6: In situ (a) $\delta^{13}$C and (b) $\delta$D versus concentration of dissolved CH$_4$ at four depths (surface water, 10 cm, 20 cm, and 30 cm) in wet areas from 2011 to 2013.**

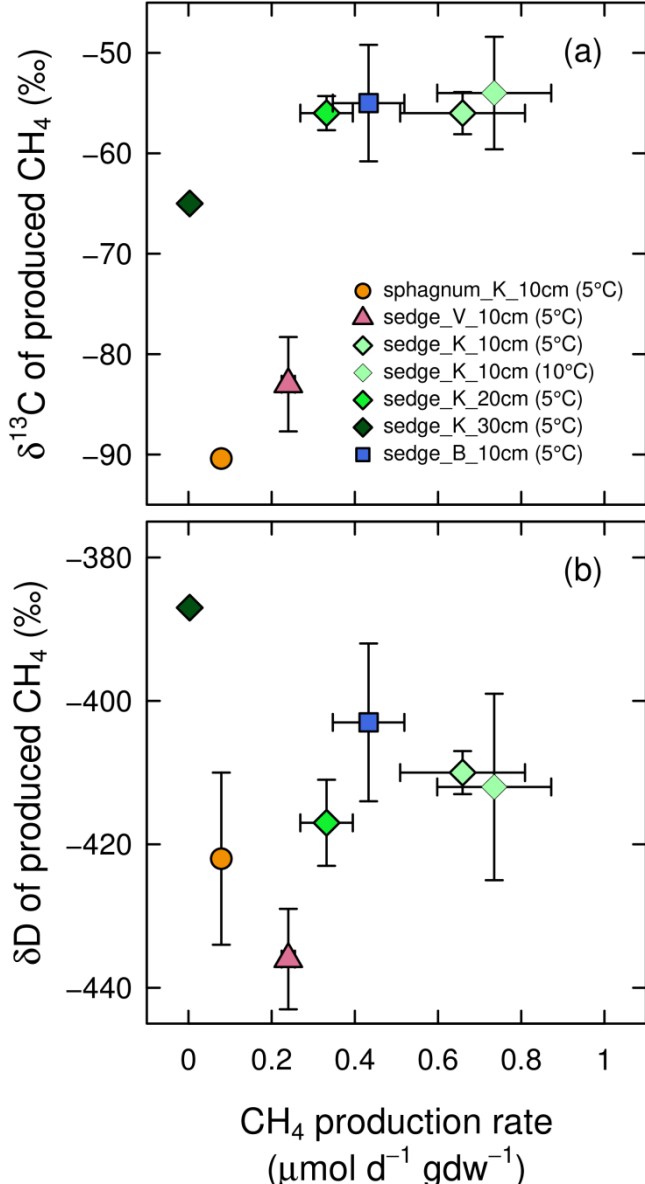

**Figure 7: (a) $\delta^{13}C$ and (b) $\delta D$ of produced $CH_4$ versus $CH_4$ production rate in the anaerobic soil incubation experiment. Production rates are shown in moles of produced $CH_4$ per day and per weight of dry soil in gram. Soil samples were collected at four observation points (sphagnum_K, sedge_V, sedge_K, and sedge_B) at three depths (10 cm, 20 cm, and 30 cm) and incubated at two temperatures (5 °C and 10 °C). These samples contain organic layers except for those collected at 30 cm. Error bars represent standard deviations.**

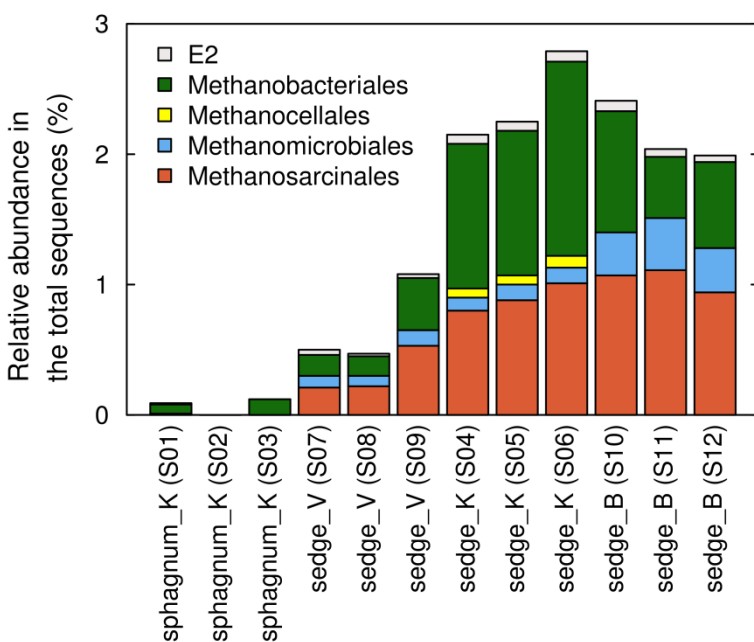

**Figure 8: Phylogenetic compositions of methanogenic archaea by order in wet areas. Soil samples (organic layers) were taken in triplicate from 10 cm depth in each wet area in July 2016. See Table S5 for detailed results.**

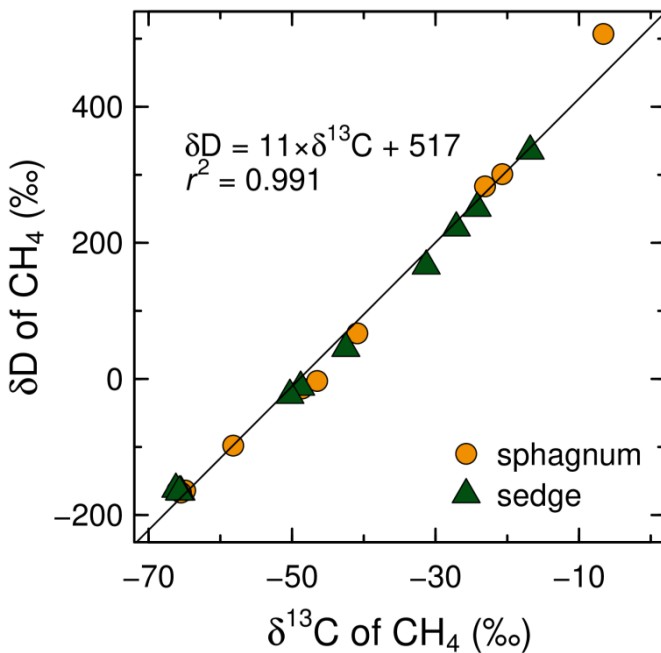

**Figure 9: Enrichment of D/H (CH$_4$) and $^{13}$C/$^{12}$C (CH$_4$) through CH$_4$ oxidation during the aerobic incubation experiment of surface organic layers from wet areas of sphagnum mosses and sedges in site K. Individual delta values of the headspace CH$_4$ from each incubated syringe and each day are shown. Initial isotopic compositions of the headspace CH$_4$ were −66‰ to −65‰ for δ$^{13}$C and −167‰ to −162‰ for δD.**