# Peer review of "Figure S2. Declines in remaining CH4 in headspace in the aerobic incubation experiment through CH4 oxidation. Surface organic layers (0-13 cm) were incubated in quadruplicate at 8 °C."

_Biogeosciences, 2018_

## Referee Comment (RC1) · Anonymous Referee #1 · 14 Nov 2018

This is an interesting article, which shows the long lasting effects - at least two years - of extreme precipitation events on CH4 emission from Arctic wetlands. It also unravels the mechanisms behind this persistence and contains an analysis of the time evolution of the CH4 production pathways over these years, showing a shift from hydrogenotopic to dominant acetoclastic methanogenesis over time.

The methodology is sound, but in particular the field methods need further clarification (detailed below). The conclusions drawn from this paper are justified, but I would have liked to see some support from field measurements of redox potentials; the field methodology description suggest that these measurements have been taken. The

same holds for active layer thickness measurements.

The authors conclude that long term water saturation is the cause of the enhanced CH4 emission, which is made plausible. However, a more extensive discussion of alternative explanations, such as increase of active layer thickness, or change of vegetation is necessary. If, for instance, the active layer measurements also suggest an increase in active layer thickness over the years following the extreme precipitation, the conclusions of the authors about the effects of water saturation should be adapted.

I have no doubt that the authors should be able to accommodate the remarks above. I recommend publication with moderate revision. In particular the Methods section needs improvement and a more in-depth discussion of the alternative causes of the long term changes in CH4 emission after extreme precipitation is necessary.

Detailed comments:

P. 3, Line 27-29: poorly readable sentence, reformulate.

P. 3, Line 30: How is 'predominantly' determined? Did you do any vegetation cover analysis?

P. 3, Line 31: What is meant by 'snapshot' measurements? What was the measurement frequency?

P. 4, Line 8: What is meant with 'principally closed'?

P. 4, Line 13: Can you give an explanation on the detection limit of your chamber measurements, for low magnitude fluxes, e.g. negative CH4 fluxes?

P. 4, Line 18: Which atmosphere? Was this ambient air or some prepared gas mixture, and what was its composition? Please clarify.

P. 4, Line 23: An ORP electrode appears to have been used for temperature measurements, but I miss the redox potential data in this article. Why have these data not been used?

P. 4, Line 26: How do you define ground surface in a sphagnum cover?

P. 5, Line 24: Calculation of the chamber fluxes: two to three values are used for calculation of the fluxes, while the field methods section says that three samples have been taken from the chambers. If in some cases only two values have been used, some of the analysis results apparently have been rejected. Clarify the reasons for rejection of samples.

P. 6, Line 15. The Chokurdagh weather station appears to be at some distance from the sites, in particular site V. Please indicate the distance.

P. 10, Line 21. In the methodology section, it is suggested that redox potential measurments were taken, however, they are not mentioned in the article. At this point, it would be very interesting to know the redox conditions over the years.

P. 11, Line 4. The 16S rRNA gene sequencing was not introduced in the Methodology section.

---

## Referee Comment (RC2) · Anonymous Referee #2 · 15 Nov 2018

Shingubara and coworkers present the multi-year effect of wet summer on CH4 emissions from riverine lowland in northeastern Siberia, which has potential to emit substantial amount of CH4. The data shown in this paper, especially carbon and hydrogen isotopic composition of CH4, contribute better understand the mechanisms of temporal variation of CH4 emission in this important region. Although I appreciate to the authors' effort to collect CH4 flux and their isotopic data in Siberia and found this is an important and worthwhile study to be done, I think the authors do not exhaust the potential of their data. Therefore, I encourage the authors to revise the manuscript. I suggest revisions on three major points as outlined below (as well as corrections of several minor points listed in the following).

[Figure]

Major points

1. If the authors want to prove that the increase of CH4 emission in 2012 and 2013 was due to reduced condition after high precipitation in 2011, the authors should show the precipitation data in the preceding years before 2009 (e.g. 2007 and 2008, if possible) to prove that low CH4 emission in 2009 and 2010 was observed under long lasting oxic condition (although there is no GWL data). By showing it, readers can convince more easily the authors' hypothesis.

2. In Figure 4 and 5, isotopic data of CH4 are shown in different colors for different year (not for each sampling site). Therefore, readers cannot see the spatial difference of these isotopic values. Please revise the figures (in the same manner as Figure S1). By doing so, the reader can judge if the difference in dD is due to spatial difference or not. In addition, are there any temporal changes in dD values at 10 cm in 2011? If there is any relationships between higher dD values and environmental factors (i.e. drop with GWL or precipitation in summer), this can be important information to understand the effect of CH4 oxidation or diffusion on variation in dD.

3. Results of phylogenic composition should be presented in the main text and as a main figure.

Minor points

Abstract

P1, L23 "soil" incubation "emitted" CH4

P1, L25 & L26 CH4 "emission"

P1, L28 "in 2011" see Major point 2

Introduction

P2, L5, Rewrite the sentence.

P2, L9-14 Referencing in the manuscript is incomplete.

P3, L8 "soil" incubation

Methods

P4, L25 When was GWL measurement conducted in each year? After every sampling? Or just one time?

P5, L3 How many soil incubation samples are prepared for each sampling point and for each initial and final measurement? Please clarify.

P5, L9-L12 If the analysis method of phylogenic composition is shown in Methods section, data (figure) should be shown as main figure (not as supplement)

P5, L15 Were the samples prepared in quadruplicate for each day of sampling? Or one sample was measured for each location and each sampling day? Please clarify.

Results

See the Major point 3.

P6, L21, Please clarify the definition of "wet area" in this manuscript.

P6, L25, Please show the thaw depth of each observation year, in addition to the averaged value.

P6, L26- See Major comment 2, please show the environmental data of several years prior to flux measurement in 2009 and 2010.

P7, L2, Again, when was GWL measurement conducted in each year? After every sampling? Or just one time? If the authors measured GWL after every sampling, it can be useful information to understand the CH4 production and oxidation processes. It may be especially true for summer 2011 when the dynamic GWL change must occur with precipitation.

P7, L11 Take out "active"

P7, L13 Take out "Interestingly"

Section 3.3 See the Major comment 2. Please show the spatial (and temporal) variations of isotopic values.

P7, L25 Please show the ranges of concentrations and dD and d13C values of CH4 in ambient air using for "in situ" dilution.

P7, L26 similarly "to what?"

P8, L9, L10, Please show statistics.

P8, L20- Please add figures showing change of d13C and dD in Figure S2.

Discussion

P8, L30, L31 Please show the ranges of CH4 flux both in this site and in the some literature.

Section 4.2 Need more reference.

P9, L15, If the authors do not show the ORP data, take out "remarkably".

P9, L26, Again, please check if these higher dD values are not associated with sampling point and sampling time.

P9, L32, Here, I recommend showing the equilibrium concentration of dissolved CH4 with atmospheric CH4, to exclude the possibility that CH4 exchange can effect on isotopic values.

P10, L1 In addition, heavy precipitation may supply O2 to surface layer of wet area.

Section 4.3 See the Major point 3. I think that the results of microbial analysis agree well with isotopic variation and, therefore, are should be shown in main text.

Concluding remarks

P11, L18-19 Add reference.

Figure 2 Please show the precipitation and temperature data in the preceding years before 2009. GWL data of sphagnum moss in 2013 seems missing.

Figure 3, Add statistical information (yearly difference) in the figure.

Figure 7, Please represent the symbols for different sampling site by different colors.

Figure 8, Are the d13C & dD data averaged value? Please clarify.

P25, L5 "in the bottom left corner"? Please rewrite.

Figure S2, Please add figures showing change in d13C and dD.

Table S2, Please show isotopic values and number of samples.

[Figure]

---

## Author Comment (AC1) · 29 Dec 2018

**Response to interactive comments from Anonymous Referee #1 (bg-2018-456)**
We would like to express our sincere gratitude to Anonymous Referee #1 for helpful comments and corrections. Our responses to specific comments (reprinted in bold) are given below.

**The methodology is sound, but in particular the field methods need further clarification (detailed below). The conclusions drawn from this paper are justified, but I would have liked to see some support from field measurements of redox potentials; the field methodology description suggest that these measurements have been taken. The same holds for active layer thickness measurements. The authors conclude that long term**
**water saturation is the cause of the enhanced $CH_4$ emission, which is made plausible. However, a more extensive discussion of alternative explanations, such as increase of active layer thickness, or change of vegetation is necessary. If, for instance, the active layer measurements also suggest an increase in active layer thickness over the years following the extreme precipitation, the conclusions of the authors about the effects of water saturation should be adapted.**
**I have no doubt that the authors should be able to accommodate the remarks above. I recommend publication with moderate revision. In particular the Methods section needs improvement and a more in-depth discussion of the alternative causes of the long term changes in $CH_4$ emission after extreme precipitation is necessary.**

*Reply:* We truly appreciate your positive suggestions.

1) Redox potential
We do not have redox potential data that can be compared among 2011 (wetting event), 2012, and 2013. We found a large artifact in redox potential value after installing the ORP electrode into soil, which lets the
atmospheric $O_2$ intrude to the soil and can increase the redox potential value. It took from several to 10 days for the redox potential to decrease and stabilize in anoxic soil.
In 2012 and 2013, we monitored temporal changes in redox potential for days or weeks after electrode installations, and obtained some redox potential data after stabilization (added as Table S6). As we have added to Sect. 4.2 in our manuscript, 'in this period, we observed redox potential values lower than −100 mV in wet areas
(Table S6), which are well below the upper limit for $CH_4$ production in soil (Conrad, 2007; Street et al., 2016). Methane production at a potential higher than −100 mV can also occur, because soil is heterogeneous and can have more reducing microsites than the rest of the bulk soil, where redox potential can be measured (Teh et al., 2005; Teh and Silver, 2006).'

2) Alternative explanations for the long term changes in $CH_4$ emission after the extreme precipitation
We have added data of interannual variation in thaw depth to the supplement as Table S1. In wet areas, we found increase in thaw depth from 2011 (22 ± 4 cm) to 2012 (25 ± 8 cm) and 2013 (35 ± 7 cm) in observations made during mid-July. We have added this information to Sect. 3.1 in the main text.
Indeed, it appears that the extreme precipitation in 2011 (Fig. 2) led to the thaw depth increase. In
addition, we did not evaluate vegetation cover quantitatively. Although we did not find drastic change in vegetation cover in the observed wet areas, abundance of sedges might have increased after the wetting event.
We have added these alternative explanations to the end of Sect. 4.2 and mentioned them in the concluding remarks in the main text.

**Detailed comments:**
**P. 3, Line 27-29: poorly readable sentence, reformulate.**
*Reply:* We have rewritten the sentence in a simpler way. For further clarity, we have also added short explanations on each name of the observation points shown in Table 1 (such as sedge_K).

**P. 3, Line 30: How is 'predominantly' determined? Did you do any vegetation cover analysis?**
*Reply:* No, this work did not analyze vegetation cover. We named a micro-relief covered by sphagnum mosses (*Sphagnum squarrosum*) as sphagnum_K, and micro-reliefs covered by sedges, especially by some cotton-sedges
(*Eriophorum angustifolium*) as sedge_V, sedge_K, sedge_B. We have taken out 'predominantly' from the manuscript and corrected Table 1 accordingly.
        On the other hand, Morozumi et al. (in review) observed specific altitude, soil moisture, and plant species along a 50 m transect in site K. They defined four vegetation classes (tree, shrub, sphagnum, and cotton-sedge), and found cotton-sedges at the lowest and wettest areas. From clustering analysis of plant species composition in
site K and a local scale (10 km × 10 km) including site K, they identified these four vegetation classes as different clusters. The vegetation types in this work, i.e. tree mound, sphagnum (wet area), and sedge (wet area) correspond to their vegetation classes (tree, sphagnum, and cotton-sedge, respectively).

**P. 3, Line 31: What is meant by 'snapshot' measurements? What was the measurement frequency?**
*Reply:* We observed volumetric water content in surface soil for three to six times at each observation point in July 2011 (Table 1). We meant by 'snapshot' that these measurements had not been conducted continuously from early July to late July but only on 1 to 3 days in July for each observation point (please refer to Table S2 for detailed observation dates). We have deleted the word 'snapshot' from the main text and added this information
to the footnote of Table 1.

**P. 4, Line 8: What is meant with 'principally closed'?**
*Reply*: We mean that 'the chamber was closed for 30 min and headspace gas was sampled at 0 min, 15 min, and
30 min after chamber closure' in most cases. We have reformulated the sentence as follows; 'The chamber was closed for 15–30 min and headspace gas was sampled for two to three times after chamber closure. In most cases, chamber was closed for 30 min and headspace gas was collected at 0 min, 15 min, and 30 min after the closure.'

**P. 4, Line 13: Can you give an explanation on the detection limit of your chamber measurements, for low magnitude fluxes, e.g. negative $CH_4$ fluxes?**
*Reply*: The detection limit of $CH_4$ flux was 0.8–2.4 mg $CH_4$ $m^{-2}$ $day^{-1}$, depending on the height of chamber headspace and conditions of the gas chromatograph in $CH_4$ concentration analyses (mentioned in Sect. 2.4). This limit also applies to negative $CH_4$ fluxes.

**P. 4, Line 18: Which atmosphere? Was this ambient air or some prepared gas mixture, and what was its composition? Please clarify.**
*Reply:* This atmosphere was collected in Chokurdakh village or our observation sites, and filtered by Molecular
Sieves 5A (1/16 pellets, FUJIFILM Wako Pure Chemical Corporation, Japan) to remove contaminants such as ammonia and n-alkanes from ethane to n-butane (carbon dioxide and water vapor are also removed significantly).

The filtered atmosphere was preserved in a 10 L aluminum bag. Its methane concentration was also analyzed by gas chromatography and found to be 2.0–4.3 ppm. We have corrected our manuscript accordingly.

**P. 4, Line 23: An ORP electrode appears to have been used for temperature measurements, but I miss the redox potential data in this article. Why have these data not been used?**
*Reply*: As we described above, we found large positive biases in redox potential after installing ORP electrodes. We obtained some redox potential data after stabilization in 2012 and 2013, and we have added these data to our manuscript as Table S6.

**P. 4, Line 26: How do you define ground surface in a sphagnum cover?**
*Reply*: In the wet area of sphagnum moss (sphagnum_K), we defined the moss surface as the origin of height, and measured water level relative to this moss surface. Thaw depth values and the depths shown with soil temperature
measurements, dissolved $CH_4$ analyses, soil incubation experiments also mean the depth relative to the moss surface if the ground surface was covered by moss. We have reformulated the sentence in our manuscript as follows; 'The water level was expressed in height relative to the ground surface or the moss surface.'

**P. 5, Line 24: Calculation of the chamber fluxes: two to three values are used for calculation of the fluxes, while the field methods section says that three samples have been taken from the chambers. If in some cases only two values have been used, some of the analysis results apparently have been rejected. Clarify the reasons for rejection of samples.**
*Reply*: We are sorry for our misleading explanation. From one chamber observation, two to three samples were
collected, and no result of sample analyses was rejected. The field methods in our manuscript have been corrected accordingly.

**P. 6, Line 15. The Chokurdagh weather station appears to be at some distance from the sites, in particular**
**site V. Please indicate the distance.**
*Reply:* The distance between Chokurdakh weather station and site V is approximately 45 km (Fig. 1). We have added this information to Sect. 2.5 in the main text.

**P. 10, Line 21. In the methodology section, it is suggested that redox potential measurments were taken, however, they are not mentioned in the article. At this point, it would be very interesting to know the redox conditions over the years.**
*Reply:* As we described above, we do not have enough reliable data for comparing redox conditions among 2011 (wetting event), 2012, and 2013. But we partly obtained some redox potential data without biases in 2012 and
2013, and we have added them to the manuscript as Table S6. We found potential values lower than −100 mV in wet areas in 2012 and 2013, which is well below the upper limit for $CH_4$ production (Conrad, 2007; Street et al., 2016).

**P. 11, Line 4. The 16S rRNA gene sequencing was not introduced in the Methodology section.**
*Reply*: We have added a description of 16S rRNA gene sequencing to the methodology section (Sect. 2.3).

---

## Author Comment (AC2) · 29 Dec 2018

**Response to interactive comments from Anonymous Referee #2 (bg-2018-456)**

We would like to express our sincere gratitude to Anonymous Referee #2 for helpful comments and corrections. Our responses to specific comments (reprinted in bold) are given below.

**Major points**

**1. If the authors want to prove that the increase of $CH_4$ emission in 2012 and 2013 was due to reduced condition after high precipitation in 2011, the authors should show the precipitation data in the preceding years before 2009 (e.g. 2007 and 2008, if possible) to prove that low $CH_4$ emission in 2009 and 2010 was**

10  **observed under long lasting oxic condition (although there is no GWL data). By showing it, readers can convince more easily the authors' hypothesis.**

*Reply*: The precipitation and air temperature data for 2007 and 2008 have been added to the manuscript (Fig. 2) from the same data source as 2009–2013 (WMO weather station 21946, GHCN-Daily). Annual precipitation was persistently low at 162–173 mm from 2007 to 2009, compared to 211–421 mm from 2010 to 2013 (in

15  hydrological year, i.e. from October in the previous year to September in the current year). This suggests dry soil conditions during our flux observations from 2009 to 2010, considering characteristically high air temperature and low precipitation in July 2010. We have revised Sect. 3.1 in our manuscript accordingly.

As we have added to the section, "Parmentier et al. (2011) reported that water level was lower in summer 2009 than the previous two summers at a tundra research station (Kytalyk) in the vicinity, approximately 30 km

20  to northwest of Chokurdakh." In addition, although we did not observe water level from 2009 to 2010 in our study area, we saw a drastic change in soil wetness conditions from 2010 to 2011, especially in sedge_V. We found no surface water even in the wettest area (sedge_V, containing some amount of cotton-sedge cover as will be described below in relation to the definition of wet area) in 2010, and we observed a high water level (10–14 cm) above the ground surface in 2011.

25

**2. In Figure 4 and 5, isotopic data of $CH_4$ are shown in different colors for different year (not for each sampling site). Therefore, readers cannot see the spatial difference of these isotopic values. Please revise the figures (in the same manner as Figure S1). By doing so, the reader can judge if the difference in dD is**

30  **due to spatial difference or not. In addition, are there any temporal changes in dD values at 10 cm in 2011? If there is any relationships between higher dD values and environmental factors (i.e. drop with GWL or**

**precipitation in summer), this can be important information to understand the effect of CH$_4$ oxidation or diffusion on variation in dD.**

*Reply*: We have added spatial information to Fig. 4 and 5 (though we also wonder if you meant corrections of Fig. 5 and 6, we hope readers can see spatial variations in δD and δ$^{13}$C of dissolved CH$_4$ from Fig. 5, and that in dissolved CH$_4$ concentration from Fig. 4). In summer 2011, three of all the four wet areas (sphagnum_K, sedge_V, and sedge_B) showed low δ$^{13}$C or high δD values apart from the convergence values (δ$^{13}$C ≅ −50‰, δD ≅ −408‰) seen in deep soil layer or under high dissolved CH$_4$ concentration (Fig. 5). In this way, it does not appear that the large variations in δ$^{13}$C and δD of dissolved CH$_4$ in 2011 were limited to one special location.

We have added individual values of water level and δD of dissolved CH$_4$ observed on each date in 2011 at 10 cm depth in wet areas to the supplement (Table S4). We found increases in water level during summer 2011. However, we could not find clear temporal change in the δD, although we only have δD data for late July in 2011. We did not find clear temporal change in the delta values of dissolved CH$_4$ in 2012 and 2013, either.

Truly, it would be our important future task to conduct detailed investigation of the temporal variation in CH$_4$ dynamics regarding precipitation and water drainage within one summer, although this study found large interannual variations in CH$_4$ flux and disolved CH$_4$ concentration, and those in isotope ratios of dissolved CH$_4$ to some extent.

**3. Results of phylogenic composition should be presented in the main text and as a main figure.**

*Reply:* We have added results of phylogenic composition to Sect. 3.4 in the main text, and moved the data figure from the supplement to the main manuscript (Fig. 8).

**Minor points**

**Abstract**

**P1, L23 "soil" incubation "emitted" CH$_4$**

*Reply:* We appreciate your corrections. We have added "soil" before "incubation" to the sentence. Instead of "emitted", we have inserted "dissolved", because we do not show any data of isotopic compositions of the emitted CH$_4$ to the atmosphere but only those of dissolved CH$_4$ for in situ observation.

**P1, L25 & L26 CH$_4$ "emission"**

*Reply:* We have corrected our manuscript accordingly.

**P1, L28 "in 2011" see Major point 2**

*Reply:* As we described above, we found no clear spatial variation and no clear temporal variation in isotopic compositions of dissolved CH$_4$.

**Introduction**

**P2, L5, Rewrite the sentence.**

*Reply:* We have rewritten the sentence.

**P2, L9-14 Referencing in the manuscript is incomplete.**

*Reply:* We have corrected our manuscript accordingly.

**P3, L8 "soil" incubation**

*Reply:* We have corrected our manuscript accordingly.

**Methods**

**P4, L25 When was GWL measurement conducted in each year? After every sampling? Or just one time?**

*Reply:* Water level was measured after most of the CH$_4$ flux observations in wet areas from 2011 to 2013. Detailed observation dates of water level are shown in Table S2. We have corrected the sentence accordingly.

**P5, L3 How many soil incubation samples are prepared for each sampling point and for each initial and final measurement? Please clarify.**

*Reply:* We prepared three soil samples for each sampling point. We collected dissolved $CH_4$ samples twice from each soil sample, and prepared three dissolved $CH_4$ samples for each of the initial and final measurements. Only for sedge_K, we prepared three replicate soil samples multiplied by four treatments of incubation (12 soil samples in total) to assess vertical variation and effect of incubation temperature. These incubation treatments were 10 cm depth at 5 °C, 10 cm depth at 10 °C, 20 cm depth at 5 °C, and 30 cm depth at 5 °C. We have added all this information to Sect. 2.3 in our manuscript.

**P5, L9-L12 If the analysis method of phylogenic composition is shown in Methods section, data (figure) should be shown as main figure (not as supplement)**

*Reply:* We have moved the data figure from the supplement to the main manuscript (as Fig. 8). We have also added detailed method of the phylogenic composition analysis to Sect. 2.3 in the main text.

**P5, L15 Were the samples prepared in quadruplicate for each day of sampling? Or one sample was measured for each location and each sampling day? Please clarify.**

*Reply:* We measured four replicate samples for each location and each sampling day. First, we prepared four replicate soil samples for each of the two sampling locations (sphagnum_K and sedge_K). Second, we collected headspace gas sample for three times (day 0, day 4 and day 8) from each incubated soil sample. We have added all this information to Sect. 2.3 in our manuscript.

**Results**

**See the Major point 3.**

*Reply:* As we described above, we have added results of phylogenic compositions to Sect. 3.4.

**P6, L21, Please clarify the definition of "wet area" in this manuscript.**

*Reply:* The definition is based on vegetation. We defined "wet area" as micro-reliefs with wetland vegetation, namely micro-reliefs covered by sphagnum mosses (*Sphagnum squarrosum*) and those by sedges, especially by some amount of cotton-sedges (*Eriophorum angustifolium*). Because wetland vegetation can be identified visually, "wet area" can be identified easily based on this definition. We found that spatial distribution of the wetland vegetation corresponded to lower elevation in microtopography and higher soil moisture from transect observation (Morozumi et al., in review). We could also confirm from Table 1 in this study that "wet areas" had higher soil moisture than tree mounds. We have rewritten the definition of wet area in Sect. 2.1 accordingly.

**P6, L25, Please show the thaw depth of each observation year, in addition to the averaged value.**

*Reply:* We have shown the thaw depth of each observation year as Table S1. In wet area, the thaw depth observed during mid-July became deeper from 2011 ($22 \pm 4$ cm) to 2012 ($25 \pm 8$ cm) and 2013 ($35 \pm 7$ cm). We have added this information to Sect. 3.1, and mentioned it in Sect. 4.2 and the concluding remarks as an alternative explanation of the multi-year effect of wetting on $CH_4$ emission.

**P6, L26- See Major comment 2, please show the environmental data of several years prior to flux measurement in 2009 and 2010.**

*Reply:* We have added precipitation and air temperature data for 2007 and 2008 to Fig. 2, and rewritten Sect. 3.1 accordingly.

**P7, L2, Again, when was GWL measurement conducted in each year? After every sampling? Or just one time? If the authors measured GWL after every sampling, it can be useful information to understand the $CH_4$ production and oxidation processes. It may be especially true for summer 2011 when the dynamic GWL change must occur with precipitation.**

*Reply:* As we described above, water level was measured after most of the $CH_4$ flux observations in wet areas from 2011 to 2013 (each observation date of water level is shown in Table S2), and we have added individual values of water level observed in wet areas on each date in 2011 to the supplement (Table S4). We found increases in water level during July 2011. However, we could not find clear temporal changes in the isotopic compositions of dissolved $CH_4$.

**P7, L11 Take out "active"**

*Reply:* We have taken out "active."

**P7, L13 Take out "Interestingly"**

*Reply:* We have taken out "Interestingly."

**Section 3.3 See the Major comment 2. Please show the spatial (and temporal) variations of isotopic values.**

*Reply:* As described above, we have added spatial information to Fig. 4 and 5. We have added data for the temporal variation of delta values at 10 cm within 2011 to the supplement (Table S4).

15

**P7, L25 Please show the ranges of concentrations and dD and d13C values of CH$_4$ in ambient air using for "in situ" dilution.**

*Reply:* We wonder if you mean the air we used for extracting dissolved CH$_4$ from water samples by headspace method. We preserved this air as a background sample for each day of dissolved CH$_4$ sampling. As a result of

20 analyzing the background samples, we obtained 2.0–4.3 ppm for CH$_4$ concentration, −53‰ to −45‰ for $\delta^{13}$C, and −168‰ to −78‰ for $\delta$D. We corrected delta values of dissolved CH$_4$ for the bias from background CH$_4$ based on mass balance. We have added these ranges to Sect. 2.2 in our manuscript.

25 **P7, L26 similarly "to what?"**

*Reply:* We intended to mention that the range of $\delta^{13}$C of dissolved CH$_4$ was similar among surface water, 10 cm depth, and 20 cm depth. We have taken out "similarly" from the sentence.

30 **P8, L9, L10, Please show statistics.**

*Reply:* With regards to the sampling depths, $CH_4$ production rate was $0.66 \pm 0.15$ µmol day$^{-1}$, $0.33 \pm 0.06$ µmol day$^{-1}$, $0.003 \pm 0.004$ µmol day$^{-1}$ for 10 cm, 20 cm, and 30 cm depths, respectively ($n = 3$ for all the depths). Difference in the rate values among the depths were significant based on Welch's ANOVA test ($p < 0.01$). Regarding the incubation temperature, production rate was $0.66 \pm 0.15$ µmol day$^{-1}$ gdw$^{-1}$ and $0.74 \pm 0.14$ µmol day$^{-1}$ gdw$^{-1}$ for 5 °C and 10 °C, respectively ($n = 3$ for both temperatures). Difference in these rate values was not significant based on *t*-test ($p > 0.5$). All these rates here were obtained for sedge_K. We have added all this information to the sentence in our manuscript.

**P8, L20- Please add figures showing change of d13C and dD in Figure S2.**

*Reply:* We have added plots of $\delta^{13}C$ and $\delta D$ to Fig. S2. As seen in these plots, both $\delta^{13}C$ and $\delta D$ increased along incubation day. Two headspace $CH_4$ samples from day 8 could not be analyzed for delta values, because the $CH_4$ concentration was low ($< 10$ ppm).

**Discussion**

**P8, L30, L31 Please show the ranges of $CH_4$ flux both in this site and in the some literature.**

*Reply:* We have reformulated the sentence as follows; "our $CH_4$ flux in wet areas (36–140 mg $CH_4$ m$^{-2}$ day$^{-1}$) was comparable to that reported for wet tundras (32–101 mg $CH_4$ m$^{-2}$ day$^{-1}$) or permafrost fens (42–147 mg $CH_4$ m$^{-2}$ day$^{-1}$) in a database across permafrost zones complied by Olefeldt et al. (2013)."

**Section 4.2 Need more reference.**

*Reply:* We have added references (Woo, 2012; Nassif and Wilson, 1975) to three sentences about hydrological processes in Sect. 4.2.

**P9, L15, If the authors do not show the ORP data, take out "remarkably".**

*Reply:* We have taken out "remarkably".

**P9, L26, Again, please check if these higher dD values are not associated with sampling point and sampling time.**

*Reply:* As we described above, we found no clear spatial variation and no clear temporal variation (Fig. 5, Table S4).

**P9, L32, Here, I recommend showing the equilibrium concentration of dissolved $CH_4$ with atmospheric $CH_4$, to exclude the possibility that $CH_4$ exchange can effect on isotopic values.**

*Reply:* We have added the following after the sentence in our manuscript. "The effect of $CH_4$ exchange between surface dissolved $CH_4$ and atmospheric $CH_4$ can be excluded, because all the dissolved $CH_4$ observed in this study was highly oversaturated ($> 0.3$ µmol $L^{-1}$, Fig. 4) compared to the equilibrium concentration of atmospheric $CH_4$ (4-5 nmol $L^{-1}$, assuming $1–10$ °C water temperature and 2 ppm atmospheric $CH_4$ concentration; Yamamoto et al., 1976)."

**P10, L1 In addition, heavy precipitation may supply $O_2$ to surface layer of wet area.**

*Reply:* We have included this thought to the sentence; "shallow layers are provided with $O_2$ from the atmosphere and precipitation."

**Section 4.3 See the Major point 3. I think that the results of microbial analysis agree well with isotopic variation and, therefore, are should be shown in main text.**

*Reply:* I truly appreciate your positive comment. We have moved the data figure of microbial analysis from the supplement to the main manuscript (as Fig. 8), and added description of the results to Sect. 3.4 in the main text. We have also modified Sect. 4.3 accordingly.

Being more confident with the interpretation, we have added the following sentence to the abstract; "delayed activation of acetoclastic methanogenesis following soil reduction could have also contributed to the enhancement of $CH_4$ production."

**Concluding remarks**

**P11, L18-19 Add reference.**

*Reply:* We have added references (Sugimoto and Wada, 1993; McCalley et al., 2014; Itoh et al., 2015).

**Figure 2 Please show the precipitation and temperature data in the preceding years before 2009. GWL data of sphagnum moss in 2013 seems missing.**

10 *Reply:* We have added precipitation and temperature data for 2007 and 2008 to the figure. As we have added to the figure caption, "water level was very low ($< -12$ cm) in the wet area of sphagnum in 2013, and could not be measured."

15 **Figure 3, Add statistical information (yearly difference) in the figure.**

*Reply:* We have added statistical information to the figure.

**Figure 7, Please represent the symbols for different sampling site by different colors.**

20 *Reply:* We have revised the figure accordingly.

**Figure 8, Are the d13C & dD data averaged value? Please clarify.**

*Reply:* The $\delta^{13}$C and $\delta$D data are individual values from each incubation syringe and each day. Nevertheless, all
25 the data points were plotted on one line. We have corrected the figure caption acccordingly (Figure 9 in our revised manuscript).

We have also corrected the ranges of both axes in the figure to include all the data points (we missed one data point with $\delta^{13}$C $= -6.6$‰ and $\delta$D $= +507$‰ in our previous manuscript).

30

**P25, L5 "in the bottom left corner"? Please rewrite.**

*Reply:* We have rewritten the sentence as follows; "initial isotopic compositions of the headspace $CH_4$ were −66‰ to −65‰ for $\delta^{13}C$ and −167‰ to −162‰ for $\delta D$."

5 **Figure S2, Please add figures showing change in d13C and dD.**

*Reply:* We have added the figures.

**Table S2, Please show isotopic values and number of samples.**

10 *Reply:* We have added isotopic values and number of samples to the table (Table S3 in our revised manuscript).

---

## Author Response (AR1)

**Response to interactive comments from Anonymous Referee #1 (bg-2018-456)**
We would like to express our sincere gratitude to Anonymous Referee #1 for helpful comments and corrections. Our responses to specific comments (reprinted in bold) are given below.

**The methodology is sound, but in particular the field methods need further clarification (detailed below). The conclusions drawn from this paper are justified, but I would have liked to see some support from field measurements of redox potentials; the field methodology description suggest that these measurements have been taken. The same holds for active layer thickness measurements. The authors conclude that long term**

10 **water saturation is the cause of the enhanced CH$_4$ emission, which is made plausible. However, a more extensive discussion of alternative explanations, such as increase of active layer thickness, or change of vegetation is necessary. If, for instance, the active layer measurements also suggest an increase in active layer thickness over the years following the extreme precipitation, the conclusions of the authors about the effects of water saturation should be adapted.**

15 **I have no doubt that the authors should be able to accommodate the remarks above. I recommend publication with moderate revision. In particular the Methods section needs improvement and a more in-depth discussion of the alternative causes of the long term changes in CH$_4$ emission after extreme precipitation is necessary.**

20 *Reply:* We truly appreciate your positive suggestions.

1) Redox potential
We do not have redox potential data that can be compared among 2011 (wetting event), 2012, and 2013. We found a large artifact in redox potential value after installing the ORP electrode into soil, which lets the

25 atmospheric O$_2$ intrude to the soil and can increase the redox potential value. It took from several to 10 days for the redox potential to decrease and stabilize in anoxic soil.
In 2012 and 2013, we monitored temporal changes in redox potential for days or weeks after electrode installations, and obtained some redox potential data after stabilization (added as Table S6). As we have added to Sect. 4.2 in our manuscript, 'in this period, we observed redox potential values lower than −100 mV in wet areas

30 (Table S6), which are well below the upper limit for CH$_4$ production in soil (Conrad, 2007; Street et al., 2016). Methane production at a potential higher than −100 mV can also occur, because soil is heterogeneous and can have more reducing microsites than the rest of the bulk soil, where redox potential can be measured (Teh et al., 2005; Teh and Silver, 2006).'

35 2) Alternative explanations for the long term changes in CH$_4$ emission after the extreme precipitation
We have added data of interannual variation in thaw depth to the supplement as Table S1. In wet areas, we found increase in thaw depth from 2011 (22 ± 4 cm) to 2012 (25 ± 8 cm) and 2013 (35 ± 7 cm) in observations made during mid-July. We have added this information to Sect. 3.1 in the main text.
Indeed, it appears that the extreme precipitation in 2011 (Fig. 2) led to the thaw depth increase. In

40 addition, we did not evaluate vegetation cover quantitatively. Although we did not find drastic change in vegetation cover in the observed wet areas, abundance of sedges might have increased after the wetting event.
We have added these alternative explanations to the end of Sect. 4.2 and mentioned them in the concluding remarks in the main text.

**Detailed comments:**

**P. 3, Line 27-29: poorly readable sentence, reformulate.**
*Reply:* We have rewritten the sentence in a simpler way. For further clarity, we have also added short explanations on each name of the observation points shown in Table 1 (such as sedge_K).

**P. 3, Line 30: How is 'predominantly' determined? Did you do any vegetation cover analysis?**
*Reply:* No, this work did not analyze vegetation cover. We named a micro-relief covered by sphagnum mosses (*Sphagnum squarrosum*) as sphagnum_K, and micro-reliefs covered by sedges, especially by some cotton-sedges
10 (*Eriophorum angustifolium*) as sedge_V, sedge_K, sedge_B. We have taken out 'predominantly' from the manuscript and corrected Table 1 accordingly.
On the other hand, Morozumi et al. (in review) observed specific altitude, soil moisture, and plant species along a 50 m transect in site K. They defined four vegetation classes (tree, shrub, sphagnum, and cotton-sedge), and found cotton-sedges at the lowest and wettest areas. From clustering analysis of plant species composition in
15 site K and a local scale (10 km × 10 km) including site K, they identified these four vegetation classes as different clusters. The vegetation types in this work, i.e. tree mound, sphagnum (wet area), and sedge (wet area) correspond to their vegetation classes (tree, sphagnum, and cotton-sedge, respectively).

20 **P. 3, Line 31: What is meant by 'snapshot' measurements? What was the measurement frequency?**
*Reply:* We observed volumetric water content in surface soil for three to six times at each observation point in July 2011 (Table 1). We meant by 'snapshot' that these measurements had not been conducted continuously from early July to late July but only on 1 to 3 days in July for each observation point (please refer to Table S2 for detailed observation dates). We have deleted the word 'snapshot' from the main text and added this information
25 to the footnote of Table 1.

**P. 4, Line 8: What is meant with 'principally closed'?**
*Reply*: We mean that 'the chamber was closed for 30 min and headspace gas was sampled at 0 min, 15 min, and
30 30 min after chamber closure' in most cases. We have reformulated the sentence as follows; 'The chamber was closed for 15–30 min and headspace gas was sampled for two to three times after chamber closure. In most cases, chamber was closed for 30 min and headspace gas was collected at 0 min, 15 min, and 30 min after the closure.'

35 **P. 4, Line 13: Can you give an explanation on the detection limit of your chamber measurements, for low magnitude fluxes, e.g. negative CH$_4$ fluxes?**
*Reply*: The detection limit of CH$_4$ flux was 0.8–2.4 mg CH$_4$ m$^{-2}$ day$^{-1}$, depending on the height of chamber headspace and conditions of the gas chromatograph in CH$_4$ concentration analyses (mentioned in Sect. 2.4). This limit also applies to negative CH$_4$ fluxes.

**P. 4, Line 18: Which atmosphere? Was this ambient air or some prepared gas mixture, and what was its composition? Please clarify.**
*Reply:* This atmosphere was collected in Chokurdakh village or our observation sites, and filtered by Molecular
45 Sieves 5A (1/16 pellets, FUJIFILM Wako Pure Chemical Corporation, Japan) to remove contaminants such as ammonia and n-alkanes from ethane to n-butane (carbon dioxide and water vapor are also removed significantly).

The filtered atmosphere was preserved in a 10 L aluminum bag. Its methane concentration was also analyzed by gas chromatography and found to be 2.0–4.3 ppm. We have corrected our manuscript accordingly.

5 **P. 4, Line 23: An ORP electrode appears to have been used for temperature measurements, but I miss the redox potential data in this article. Why have these data not been used?**
*Reply*: As we described above, we found large positive biases in redox potential after installing ORP electrodes. We obtained some redox potential data after stabilization in 2012 and 2013, and we have added these data to our manuscript as Table S6.

**P. 4, Line 26: How do you define ground surface in a sphagnum cover?**
*Reply*: In the wet area of sphagnum moss (sphagnum_K), we defined the moss surface as the origin of height, and measured water level relative to this moss surface. Thaw depth values and the depths shown with soil temperature
15 measurements, dissolved $CH_4$ analyses, soil incubation experiments also mean the depth relative to the moss surface if the ground surface was covered by moss. We have reformulated the sentence in our manuscript as follows; 'The water level was expressed in height relative to the ground surface or the moss surface.'

20 **P. 5, Line 24: Calculation of the chamber fluxes: two to three values are used for calculation of the fluxes, while the field methods section says that three samples have been taken from the chambers. If in some cases only two values have been used, some of the analysis results apparently have been rejected. Clarify the reasons for rejection of samples.**
*Reply*: We are sorry for our misleading explanation. From one chamber observation, two to three samples were
25 collected, and no result of sample analyses was rejected. The field methods in our manuscript have been corrected accordingly.

**P. 6, Line 15. The Chokurdagh weather station appears to be at some distance from the sites, in particular**
30 **site V. Please indicate the distance.**
*Reply:* The distance between Chokurdakh weather station and site V is approximately 45 km (Fig. 1). We have added this information to Sect. 2.5 in the main text.

35 **P. 10, Line 21. In the methodology section, it is suggested that redox potential measurments were taken, however, they are not mentioned in the article. At this point, it would be very interesting to know the redox conditions over the years.**
*Reply:* As we described above, we do not have enough reliable data for comparing redox conditions among 2011 (wetting event), 2012, and 2013. But we partly obtained some redox potential data without biases in 2012 and
40 2013, and we have added them to the manuscript as Table S6. We found potential values lower than −100 mV in wet areas in 2012 and 2013, which is well below the upper limit for $CH_4$ production (Conrad, 2007; Street et al., 2016).

45 **P. 11, Line 4. The 16S rRNA gene sequencing was not introduced in the Methodology section.**
*Reply*: We have added a description of 16S rRNA gene sequencing to the methodology section (Sect. 2.3).

**Response to interactive comments from Anonymous Referee #2 (bg-2018-456)**

We would like to express our sincere gratitude to Anonymous Referee #2 for helpful comments and corrections. Our responses to specific comments (reprinted in bold) are given below.

**Major points**

**1. If the authors want to prove that the increase of CH$_4$ emission in 2012 and 2013 was due to reduced condition after high precipitation in 2011, the authors should show the precipitation data in the preceding years before 2009 (e.g. 2007 and 2008, if possible) to prove that low CH$_4$ emission in 2009 and 2010 was**

10 **observed under long lasting oxic condition (although there is no GWL data). By showing it, readers can convince more easily the authors' hypothesis.**

*Reply*: The precipitation and air temperature data for 2007 and 2008 have been added to the manuscript (Fig. 2) from the same data source as 2009–2013 (WMO weather station 21946, GHCN-Daily). Annual precipitation was persistently low at 162–173 mm from 2007 to 2009, compared to 211–421 mm from 2010 to 2013 (in

15 hydrological year, i.e. from October in the previous year to September in the current year). This suggests dry soil conditions during our flux observations from 2009 to 2010, considering characteristically high air temperature and low precipitation in July 2010. We have revised Sect. 3.1 in our manuscript accordingly.

As we have added to the section, "Parmentier et al. (2011) reported that water level was lower in summer 2009 than the previous two summers at a tundra research station (Kytalyk) in the vicinity, approximately 30 km

20 to northwest of Chokurdakh." In addition, although we did not observe water level from 2009 to 2010 in our study area, we saw a drastic change in soil wetness conditions from 2010 to 2011, especially in sedge_V. We found no surface water even in the wettest area (sedge_V, containing some amount of cotton-sedge cover as will be described below in relation to the definition of wet area) in 2010, and we observed a high water level (10–14 cm) above the ground surface in 2011.

**2. In Figure 4 and 5, isotopic data of CH$_4$ are shown in different colors for different year (not for each sampling site). Therefore, readers cannot see the spatial difference of these isotopic values. Please revise the figures (in the same manner as Figure S1). By doing so, the reader can judge if the difference in dD is**

30 **due to spatial difference or not. In addition, are there any temporal changes in dD values at 10 cm in 2011? If there is any relationships between higher dD values and environmental factors (i.e. drop with GWL or**

**precipitation in summer), this can be important information to understand the effect of CH$_4$ oxidation or diffusion on variation in dD.**

*Reply*: We have added spatial information to Fig. 4 and 5 (though we also wonder if you meant corrections of Fig. 5 and 6, we hope readers can see spatial variations in δD and δ$^{13}$C of dissolved CH$_4$ from Fig. 5, and that in dissolved CH$_4$ concentration from Fig. 4). In summer 2011, three of all the four wet areas (sphagnum_K, sedge_V, and sedge_B) showed low δ$^{13}$C or high δD values apart from the convergence values (δ$^{13}$C $\cong$ −50‰, δD $\cong$ −408‰) seen in deep soil layer or under high dissolved CH$_4$ concentration (Fig. 5). In this way, it does not appear that the large variations in δ$^{13}$C and δD of dissolved CH$_4$ in 2011 were limited to one special location.

We have added individual values of water level and δD of dissolved CH$_4$ observed on each date in 2011 at 10 cm depth in wet areas to the supplement (Table S4). We found increases in water level during summer 2011. However, we could not find clear temporal change in the δD, although we only have δD data for late July in 2011. We did not find clear temporal change in the delta values of dissolved CH$_4$ in 2012 and 2013, either.

Truly, it would be our important future task to conduct detailed investigation of the temporal variation in CH$_4$ dynamics regarding precipitation and water drainage within one summer, although this study found large interannual variations in CH$_4$ flux and disolved CH$_4$ concentration, and those in isotope ratios of dissolved CH$_4$ to some extent.

**3. Results of phylogenic composition should be presented in the main text and as a main figure.**

*Reply:* We have added results of phylogenic composition to Sect. 3.4 in the main text, and moved the data figure from the supplement to the main manuscript (Fig. 8).

**Minor points**

**Abstract**

**P1, L23 "soil" incubation "emitted" CH$_4$**

*Reply:* We appreciate your corrections. We have added "soil" before "incubation" to the sentence. Instead of "emitted", we have inserted "dissolved", because we do not show any data of isotopic compositions of the emitted CH$_4$ to the atmosphere but only those of dissolved CH$_4$ for in situ observation.

**P1, L25 & L26 CH$_4$ "emission"**

*Reply:* We have corrected our manuscript accordingly.

**P1, L28 "in 2011" see Major point 2**

*Reply:* As we described above, we found no clear spatial variation and no clear temporal variation in isotopic compositions of dissolved CH$_4$.

**Introduction**

**P2, L5, Rewrite the sentence.**

*Reply:* We have rewritten the sentence.

**P2, L9-14 Referencing in the manuscript is incomplete.**

*Reply:* We have corrected our manuscript accordingly.

**P3, L8 "soil" incubation**

*Reply:* We have corrected our manuscript accordingly.

**Methods**

**P4, L25 When was GWL measurement conducted in each year? After every sampling? Or just one time?**

*Reply:* Water level was measured after most of the CH$_4$ flux observations in wet areas from 2011 to 2013. Detailed observation dates of water level are shown in Table S2. We have corrected the sentence accordingly.

**P5, L3 How many soil incubation samples are prepared for each sampling point and for each initial and final measurement? Please clarify.**

*Reply:* We prepared three soil samples for each sampling point. We collected dissolved $CH_4$ samples twice from each soil sample, and prepared three dissolved $CH_4$ samples for each of the initial and final measurements. Only for sedge_K, we prepared three replicate soil samples multiplied by four treatments of incubation (12 soil samples in total) to assess vertical variation and effect of incubation temperature. These incubation treatments were 10 cm depth at 5 °C, 10 cm depth at 10 °C, 20 cm depth at 5 °C, and 30 cm depth at 5 °C. We have added all this information to Sect. 2.3 in our manuscript.

**P5, L9-L12 If the analysis method of phylogenic composition is shown in Methods section, data (figure) should be shown as main figure (not as supplement)**

*Reply:* We have moved the data figure from the supplement to the main manuscript (as Fig. 8). We have also added detailed method of the phylogenic composition analysis to Sect. 2.3 in the main text.

**P5, L15 Were the samples prepared in quadruplicate for each day of sampling? Or one sample was measured for each location and each sampling day? Please clarify.**

*Reply:* We measured four replicate samples for each location and each sampling day. First, we prepared four replicate soil samples for each of the two sampling locations (sphagnum_K and sedge_K). Second, we collected headspace gas sample for three times (day 0, day 4 and day 8) from each incubated soil sample. We have added all this information to Sect. 2.3 in our manuscript.

**Results**

**See the Major point 3.**

*Reply:* As we described above, we have added results of phylogenic compositions to Sect. 3.4.

**P6, L21, Please clarify the definition of "wet area" in this manuscript.**

*Reply:* The definition is based on vegetation. We defined "wet area" as micro-reliefs with wetland vegetation, namely micro-reliefs covered by sphagnum mosses (*Sphagnum squarrosum*) and those by sedges, especially by some amount of cotton-sedges (*Eriophorum angustifolium*). Because wetland vegetation can be identified visually, "wet area" can be identified easily based on this definition. We found that spatial distribution of the wetland vegetation corresponded to lower elevation in microtopography and higher soil moisture from transect observation (Morozumi et al., in review). We could also confirm from Table 1 in this study that "wet areas" had higher soil moisture than tree mounds. We have rewritten the definition of wet area in Sect. 2.1 accordingly.

**P6, L25, Please show the thaw depth of each observation year, in addition to the averaged value.**

*Reply:* We have shown the thaw depth of each observation year as Table S1. In wet area, the thaw depth observed during mid-July became deeper from 2011 ($22 \pm 4$ cm) to 2012 ($25 \pm 8$ cm) and 2013 ($35 \pm 7$ cm). We have added this information to Sect. 3.1, and mentioned it in Sect. 4.2 and the concluding remarks as an alternative explanation of the multi-year effect of wetting on $CH_4$ emission.

**P6, L26- See Major comment 2, please show the environmental data of several years prior to flux measurement in 2009 and 2010.**

*Reply:* We have added precipitation and air temperature data for 2007 and 2008 to Fig. 2, and rewritten Sect. 3.1 accordingly.

**P7, L2, Again, when was GWL measurement conducted in each year? After every sampling? Or just one time? If the authors measured GWL after every sampling, it can be useful information to understand the $CH_4$ production and oxidation processes. It may be especially true for summer 2011 when the dynamic GWL change must occur with precipitation.**

*Reply:* As we described above, water level was measured after most of the $CH_4$ flux observations in wet areas from 2011 to 2013 (each observation date of water level is shown in Table S2), and we have added individual values of water level observed in wet areas on each date in 2011 to the supplement (Table S4). We found increases in water level during July 2011. However, we could not find clear temporal changes in the isotopic compositions of dissolved $CH_4$.

**P7, L11 Take out "active"**

*Reply:* We have taken out "active."

**P7, L13 Take out "Interestingly"**

*Reply:* We have taken out "Interestingly."

**Section 3.3 See the Major comment 2. Please show the spatial (and temporal) variations of isotopic values.**

*Reply:* As described above, we have added spatial information to Fig. 4 and 5. We have added data for the temporal variation of delta values at 10 cm within 2011 to the supplement (Table S4).

**P7, L25 Please show the ranges of concentrations and dD and d13C values of CH$_4$ in ambient air using for "in situ" dilution.**

*Reply:* We wonder if you mean the air we used for extracting dissolved CH$_4$ from water samples by headspace method. We preserved this air as a background sample for each day of dissolved CH$_4$ sampling. As a result of

20     analyzing the background samples, we obtained 2.0–4.3 ppm for CH$_4$ concentration, −53‰ to −45‰ for $\delta^{13}$C, and −168‰ to −78‰ for $\delta$D. We corrected delta values of dissolved CH$_4$ for the bias from background CH$_4$ based on mass balance. We have added these ranges to Sect. 2.2 in our manuscript.

25     **P7, L26 similarly "to what?"**

*Reply:* We intended to mention that the range of $\delta^{13}$C of dissolved CH$_4$ was similar among surface water, 10 cm depth, and 20 cm depth. We have taken out "similarly" from the sentence.

30     **P8, L9, L10, Please show statistics.**

*Reply:* With regards to the sampling depths, CH$_4$ production rate was $0.66 \pm 0.15$ µmol day$^{-1}$, $0.33 \pm 0.06$ µmol day$^{-1}$, $0.003 \pm 0.004$ µmol day$^{-1}$ for 10 cm, 20 cm, and 30 cm depths, respectively ($n = 3$ for all the depths). Difference in the rate values among the depths were significant based on Welch's ANOVA test ($p < 0.01$). Regarding the incubation temperature, production rate was $0.66 \pm 0.15$ µmol day$^{-1}$ gdw$^{-1}$ and $0.74 \pm 0.14$ µmol day$^{-1}$ gdw$^{-1}$ for 5 °C and 10 °C, respectively ($n = 3$ for both temperatures). Difference in these rate values was not significant based on *t*-test ($p > 0.5$). All these rates here were obtained for sedge_K. We have added all this information to the sentence in our manuscript.

**P8, L20- Please add figures showing change of d13C and dD in Figure S2.**

*Reply:* We have added plots of $\delta^{13}$C and $\delta$D to Fig. S2. As seen in these plots, both $\delta^{13}$C and $\delta$D increased along incubation day. Two headspace CH$_4$ samples from day 8 could not be analyzed for delta values, because the CH$_4$ concentration was low ($< 10$ ppm).

**Discussion**

**P8, L30, L31 Please show the ranges of CH$_4$ flux both in this site and in the some literature.**

*Reply:* We have reformulated the sentence as follows; "our CH$_4$ flux in wet areas (36–140 mg CH$_4$ m$^{-2}$ day$^{-1}$) was comparable to that reported for wet tundras (32–101 mg CH$_4$ m$^{-2}$ day$^{-1}$) or permafrost fens (42–147 mg CH$_4$ m$^{-2}$ day$^{-1}$) in a database across permafrost zones complied by Olefeldt et al. (2013)."

**Section 4.2 Need more reference.**

*Reply:* We have added references (Woo, 2012; Nassif and Wilson, 1975) to three sentences about hydrological processes in Sect. 4.2.

**P9, L15, If the authors do not show the ORP data, take out "remarkably".**

*Reply:* We have taken out "remarkably".

**P9, L26, Again, please check if these higher dD values are not associated with sampling point and sampling time.**

*Reply:* As we described above, we found no clear spatial variation and no clear temporal variation (Fig. 5, Table S4).

**P9, L32, Here, I recommend showing the equilibrium concentration of dissolved $CH_4$ with atmospheric $CH_4$, to exclude the possibility that $CH_4$ exchange can effect on isotopic values.**

*Reply:* We have added the following after the sentence in our manuscript. "The effect of $CH_4$ exchange between surface dissolved $CH_4$ and atmospheric $CH_4$ can be excluded, because all the dissolved $CH_4$ observed in this study was highly oversaturated ($> 0.3$ μmol $L^{-1}$, Fig. 4) compared to the equilibrium concentration of atmospheric $CH_4$ (4-5 nmol $L^{-1}$, assuming $1-10$ °C water temperature and 2 ppm atmospheric $CH_4$ concentration; Yamamoto et al., 1976)."

**P10, L1 In addition, heavy precipitation may supply $O_2$ to surface layer of wet area.**

*Reply:* We have included this thought to the sentence; "shallow layers are provided with $O_2$ from the atmosphere and precipitation."

**Section 4.3 See the Major point 3. I think that the results of microbial analysis agree well with isotopic variation and, therefore, are should be shown in main text.**

*Reply:* I truly appreciate your positive comment. We have moved the data figure of microbial analysis from the supplement to the main manuscript (as Fig. 8), and added description of the results to Sect. 3.4 in the main text. We have also modified Sect. 4.3 accordingly.

Being more confident with the interpretation, we have added the following sentence to the abstract; "delayed activation of acetoclastic methanogenesis following soil reduction could have also contributed to the enhancement of $CH_4$ production."

**Concluding remarks**

**P11, L18-19 Add reference.**

*Reply:* We have added references (Sugimoto and Wada, 1993; McCalley et al., 2014; Itoh et al., 2015).

**Figure 2 Please show the precipitation and temperature data in the preceding years before 2009. GWL data of sphagnum moss in 2013 seems missing.**

10 *Reply:* We have added precipitation and temperature data for 2007 and 2008 to the figure. As we have added to the figure caption, "water level was very low ($< -12$ cm) in the wet area of sphagnum in 2013, and could not be measured."

15 **Figure 3, Add statistical information (yearly difference) in the figure.**

*Reply:* We have added statistical information to the figure.

**Figure 7, Please represent the symbols for different sampling site by different colors.**

20 *Reply:* We have revised the figure accordingly.

**Figure 8, Are the d13C & dD data averaged value? Please clarify.**

*Reply:* The $\delta^{13}$C and $\delta$D data are individual values from each incubation syringe and each day. Nevertheless, all
25 the data points were plotted on one line. We have corrected the figure caption acccordingly (Figure 9 in our revised manuscript).

We have also corrected the ranges of both axes in the figure to include all the data points (we missed one data point with $\delta^{13}$C = $-6.6$‰ and $\delta$D = $+507$‰ in our previous manuscript).

**P25, L5 "in the bottom left corner"? Please rewrite.**

*Reply:* We have rewritten the sentence as follows; "initial isotopic compositions of the headspace $CH_4$ were −66‰ to −65‰ for $\delta^{13}C$ and −167‰ to −162‰ for $\delta D$."

5 **Figure S2, Please add figures showing change in d13C and dD.**

*Reply:* We have added the figures.

**Table S2, Please show isotopic values and number of samples.**

10 *Reply:* We have added isotopic values and number of samples to the table (Table S3 in our revised manuscript).

[revised manuscript text omitted]

Microbial community analysis by amplicon sequence of 16S rRNA gene was applied to soil samples at 10 cm depth at the same locations as the $CH_4$ production incubation experiment (Fig. S3, Table S3). Soils with high rates of $CH_4$ production and high $\delta^{13}C$ of $CH_4$ produced in incubation (sedge_K and sedge_B as in Fig. 7) had higher proportion of acetoclastic methanogens in the order Methanosarcinales than those with low $CH_4$ production rates and low $\delta^{13}C$ of produced $CH_4$ (sphagnum_K and sedge_V). This supports the interpretation that the ratio of acetoclastic to hydrogenotrophic methanogenesis controlled the $\delta^{13}C$ of produced $CH_4$ in incubation.

[revised manuscript text omitted]

..........................改ページ..........................
.
.
...

**Figure S2**. Temporal changes in (a) concentration, (b) $\delta^{13}$C, and (c) $\delta$D of the remaining headspace CH$_4$ in the soil incubation experiment for CH$_4$ production. Surface organic layers (0-13 cm) from wet areas (sphagnum_K and sedge_K) were incubated in quadruplicate at 8 °C.

**Table S1**. Thaw depths in tree mound and wet area observed along with $CH_4$ flux from 2010 to 2013. Averaged values and ranges are shown for each vegetation type and each year. Standard deviations are represented when $n \geq 3$. See Table S2 for each observation date.

| Year | Thaw depth (cm) | | | |
|---|---|---|---|---|
| | Tree mound | | Wet area | |
| | mean | range | mean | range |
| 2010 | 23 ($n = 2$) | 21–25 (Jul 20–21) | 30 ($n = 2$) | 29–30 (Jul 20–21) |
| 2011 | 23 ± 6 ($n = 6$) | 14–30 (Jul 9–30) | 22 ± 4 ($n = 9$) | 15–28 (Jul 9–21) |
| 2012 | 21 ± 4 ($n = 11$) | 16–27 (Jul 3 – Aug 9) | 35 ± 14 ($n = 24$)[a] | 9–57 (Jul 3 – Aug 9)[a] |
| 2013 | 20 ± 3 ($n = 3$) | 18–23 (Jul 15 – Aug 2) | 40 ± 9 ($n = 20$)[b] | 23–58 (Jul 11 – Aug 2)[b] |

[a] During Jul 8–20, 2012 in wet area, the mean value was 25 ± 8 cm, and the range was 9–37 cm.

[b] During Jul 11–18, 2013 in wet area, the mean value was 35 ± 7 cm, and the range was 23–46 cm.

表の書式変更

**Table S2.** Averaged CH$_4$ flux (in mg CH$_4$ m$^{-2}$ day$^{-1}$) over each observation point and each year (2009–2013). Standard deviations are shown in case of $n \geq 3$. Dates of the flux observation are indicated in parenthesis. Superscripts represent observed environmental variables on each day: a) soil temperature (2009–2013), b) thaw depth (2010–2013), c) water level (2011–2013), and d) volumetric water content in surface soil (2011).

| Observation points | Year | | | | |
|---|---|---|---|---|---|
| | 2009 | 2010 | 2011 | 2012 | 2013 |
| tree mound_V | – | 0 (Jul 16) | −1 ± 2 (Jul 23[b], 29[abd]) | 0 (Aug 7[ab]) | 0 (Aug 2[ab]) |
| sedge_V (wet area) | 3 ± 3 (Jul 23) | 2 (Jul 16[a]) | 179 ± 66 (Jul 23[c], 29[acd]) | 46 (Aug 7[abc]) | 106 ± 21 (Aug 2[abc]) |
| tree mound_K | 0 (Jul 22) | −1 ± 3 (Jul 21[ab]) | 0 (Jul 15[bd], 18[bd]) | 0 (Jul 3[ab], 8[a], 12[ab], 24[ab]; Aug 2[ab], 6[ab]) | 1 (Jul 15[ab]) |
| sphagnum_K (wet area) | – | 1 ± 1 (Jul 21[ab]) | 43 ± 31 (Jul 11[bd], 17[b], 18[bcd], 21[bcd]) | 3 ± 3 (Jul 3[abc], 8[abc], 12[abc], 24[abc]; Aug 2[abc], 6[abc]) | 102 ± 4 (Jul 11[ab], 18[ab], 25[ab], 31[ab]) |
| sedge_K (wet area) | 26 ± 24 (Jul 22[a]) | – | 28 ± 4 (Jul 11[bcd], 17[bc], 18[bcd], 21[abcd]) | 83 ± 30 (Jul 3[abc], 8[abc], 12[abc], 20[abc], 21[a], 24[ab]; Aug 2[abc], 6[abc]) | 111 ± 63 (Jul 11[abc], 18[abc], 25[abc], 31[abc]) |
| tree mound_B | – | 0 (Jul 20[b]) | 0 (Jul 9[bd], 30[bd]) | −1 ± 1 (Jul 13[ab], Aug 9[ab]) | 0 (Jul 16[ab]) |
| sedge_B (wet area) | 79 ± 80 (Jul 25) | 98 ± 84 (Jul 20[b]) | 151 ± 58 (Jul 9[bcd], 30[acd]) | 131 ± 58 (Jul 13[ab], Aug 9[abc]) | 286 ± 49 (Jul 16[abc]) |

**Table S3.** Concentration, δ¹³C, and δD of dissolved CH$_4$ in surface water and soil pore water averaged over each wet area and each year. Standard deviations are shown in case of $n \geq 3$.

| Wet area | Depth | Concentration (μmol CH$_4$ L$^{-1}$) | | | δ¹³C (‰) | | | δD (‰) | | |
|---|---|---|---|---|---|---|---|---|---|---|
| | | 2011 | 2012 | 2013 | 2011 | 2012 | 2013 | 2011 | 2012 | 2013 |
| sphagnum_K | surface water | — | — | — | — | — | — | — | — | — |
| | 10 cm | 6 ± 3 (n = 6) | 318 (n = 1) | — | −56 ± 9 (n = 6) | −48 (n = 1) | — | −380 ± 23 (n = 6) | −408 (n = 1) | — |
| | 20 cm | — | 176 ± 125 (n = 3) | 399 ± 68 (n = 3) | — | −55 ± 10 (n = 3) | −47 ± 2 (n = 3) | — | −412 ± 4 (n = 3) | −412 ± 3 (n = 3) |
| | 30 cm | — | 266 (n = 2) | 351 (n = 1) | — | −49 (n = 2) | −50 (n = 1) | — | −407 (n = 2) | −406 (n = 1) |
| sedge_V | surface water | 3 ± 2 (n = 3) | 6 (n = 2) | 105 (n = 2) | −58.1 ± 0.2 (n = 3) | −56 (n = 2) | −60 (n = 2) | −378 ± 7 (n = 3) | −383 (n = 1) | −414 (n = 2) |
| | 10 cm | 30 ± 17 (n = 3) | 150 (n = 1) | 249 (n = 1) | −59 (n = 2) | −51 (n = 1) | −46 (n = 1) | −405 (n = 2) | −409 (n = 1) | −415 (n = 1) |
| | 20 cm | — | 52 (n = 1) | 168 (n = 1) | — | −53 (n = 1) | −51 (n = 1) | — | −405 (n = 1) | −416 (n = 1) |
| | 30 cm | — | — | — | — | — | — | — | — | — |
| sedge_K | surface water | 12 ± 14 (n = 4) | 8 ± 8 (n = 11) | 74 ± 27 (n = 6) | −46 ± 4 (n = 4) | −51 ± 9 (n = 10) | −45 ± 3 (n = 6) | −378 ± 27 (n = 4) | −349 ± 30 (n = 4) | −388 ± 11 (n = 6) |
| | 10 cm | 53 ± 31 (n = 6) | 244 ± 143 (n = 7) | 88 ± 27 (n = 5) | −50 ± 1 (n = 6) | −52 ± 3 (n = 7) | −49 ± 4 (n = 5) | −412 ± 4 (n = 6) | −405 ± 21 (n = 7) | −392 ± 11 (n = 5) |
| | 20 cm | — | 368 ± 59 (n = 3) | 431 ± 123 (n = 5) | — | −46 ± 1 (n = 3) | −54 ± 10 (n = 5) | — | −412 ± 1 (n = 3) | −407 ± 4 (n = 5) |
| | 30 cm | — | 519 (n = 2) | 450 ± 55 (n = 3) | — | −50 (n = 2) | −49 ± 2 (n = 3) | — | −412 (n = 2) | −400 ± 2 (n = 3) |
| sedge_B | surface water | 14 (n = 2) | 8 ± 14 (n = 3) | 4 (n = 2) | −40 (n = 2) | −56 (n = 1) | −48 (n = 2) | −320 (n = 2) | −392 (n = 1) | −350 (n = 1) |
| | 10 cm | 21 (n = 2) | 380 (n = 1) | 392 (n = 1) | −47 (n = 2) | −53 (n = 1) | −51 (n = 1) | −360 (n = 2) | −405 (n = 1) | −404 (n = 1) |
| | 20 cm | — | 351 (n = 2) | 322 (n = 1) | — | −54 (n = 2) | −53 (n = 1) | — | −404 (n = 2) | −406 (n = 1) |
| | 30 cm | — | — | 194 (n = 1) | — | — | −53 (n = 1) | — | — | −405 (n = 1) |

**Table S4**. Individual values of water level, dissolved $CH_4$ concentration (10 cm depth), and $\delta^{13}C$ and $\delta D$ of dissolved $CH_4$ (10 cm depth) observed in each wet area on each date in 2011. Duplicated data are shown in some cases. Although water level increased during July 2011, clear temporal change was not found in the delta values.

| Observation point | Date in 2011 | Water level (cm) | Dissolved $CH_4$ concentration ($\mu$mol L$^{-1}$) | $\delta^{13}C$ of dissolved $CH_4$ (‰) | $\delta D$ of dissolved $CH_4$ (‰) |
|---|---|---|---|---|---|
| sphagnum_K | Jul 17 | — (< 0) | 6, 8 | −43, −58 | −386, −394 |
| | Jul 18 | −10 | 2, 2 | −56, −51 | −363, −341 |
| | Jul 21 | −8 | 8, 9 | −66, −65 | −398, −395 |
| sedge_V | Jul 23 | 10 | 16 | −58 | −409 |
| | Jul 29 | 14 | 49, 26 | −60 | −401 |
| sedge_K | Jul 11 | 5 | − | − | − |
| | Jul 17 | 15 | 27, 37 | −50, −52 | −407, −408 |
| | Jul 18 | 15 | 32, 35 | −51, −50 | −413, −414 |
| | Jul 21 | 15 | 88, 97 | −49, −49 | −415, −415 |
| sedge_B | Jul 9 | 6 | − | − | − |
| | Jul 30 | 11 | 17, 26 | −45, −49 | −350, −369 |

表の書式変更

**Table S5.** Phylogenic composition of methanogenic archaea in wet areas. Soils (organic layers) were sampled in July 2016 from 10 cm depth in the same wet areas as the CH$_4$ production incubation experiment in triplicate. Then, microbial communities in the samples were analyzed by amplicon sequencing of 16S rRNA gene. See Fig. 8 for a plot by the level of orders.

| Order | Family | Genus | Relative abundance in the total sequences (%) | | | | | | | | | | | |
|---|---|---|---|---|---|---|---|---|---|---|---|---|---|---|
| | | | sphagnu m_K (S01) | sphagnu m_K (S02) | sphagnu m_K (S03) | sedge _V (S07) | sedge _V (S08) | sedge _V (S09) | sedge _K (S04) | sedge _K (S05) | sedge _K (S06) | sedge _B (S10) | sedge _B (S11) | sedge _B (S12) |
| Methanosarci nales | Methanosarcinacea e | *Methanosarcina* | 0.01 | 0.00 | 0.00 | 0.21 | 0.09 | 0.47 | 0.06 | 0.08 | 0.08 | 0.25 | 0.20 | 0.16 |
| | Methanosaetaceae | *Methanosaeta* | 0.00 | 0.00 | 0.00 | 0.00 | 0.13 | 0.06 | 0.74 | 0.80 | 0.93 | 0.82 | 0.91 | 0.78 |
| Methanomicr obiales | Methanoregulaceae | *Candidatus* Methanoregula | 0.00 | 0.00 | 0.00 | 0.09 | 0.07 | 0.10 | 0.10 | 0.10 | 0.11 | 0.23 | 0.32 | 0.28 |
| | Methanospirillacea e | *Methanospirillum* | 0.00 | 0.00 | 0.00 | 0.00 | 0.00 | 0.01 | 0.00 | 0.00 | 0.00 | 0.02 | 0.00 | 0.01 |
| | Methanoregulaceae | Other Methanoregulaceae | 0.00 | 0.00 | 0.00 | 0.00 | 0.01 | 0.01 | 0.00 | 0.02 | 0.01 | 0.08 | 0.08 | 0.05 |
| Methanocella les | | Methanocellales | 0.00 | 0.00 | 0.00 | 0.00 | 0.00 | 0.00 | 0.07 | 0.07 | 0.09 | 0.00 | 0.00 | 0.00 |
| Methanobacte riales | Methanobacteriace ae | *Methanobacterium* | 0.07 | 0.00 | 0.12 | 0.16 | 0.15 | 0.40 | 1.11 | 1.11 | 1.49 | 0.93 | 0.47 | 0.66 |
| E2 | [Methanomassiliic occaceae] | [Methanomassiliic occaceae] | 0.01 | 0.00 | 0.00 | 0.04 | 0.02 | 0.03 | 0.07 | 0.07 | 0.08 | 0.08 | 0.06 | 0.05 |

[Figure]

： 10 mm, 下： 23.6 mm, 幅： 210 mm, 高
さ： 240 mm

**Table S6**. Redox potential observed in summers 2012 and 2013 by ORP meter (RM-20P or RM-30P, DKK-TOA Corporation, Japan) connected with an ORP electrode (PST-2739C). Measurement accuracy of the ORP meter is ±10 mV. Redox potential value was accepted when the potential stabilized after installing the ORP electrode into the soil.

| Observation points | Depth (cm) | Redox potential (mV versus the normal hydrogen electrode) | |
| --- | --- | --- | --- |
| | | 2012 | 2013 |
| tree mound_K | 10 | 608 to 643 | 478 to 482 |
| | 20 | 627 to 631 | — |
| sphagnum_K (wet area) | 10 | −183 to 814 | 547 to 617 |
| | 20 | −129 | — |
| sedge_K (wet area) | 10 | −177 to −121 | −114 to −69 |
| | 20 | −250 to −78 | −223 to −194 |
| | 30 | −152 to −118 | — |
| sedge_B (wet area) | 10 | — | −113 to −102 |